# Beyond Model Collapse: Scaling Up with Synthesized Data Requires Verification

**Yunzhen Feng**[1,2,*,‖] **Elvis Dohmatob**[1,3,4,*] **Pu Yang**[5,*] **Francois Charton**[1] **Julia Kempe**[1,2]

[1]Meta FAIR  [2]New York University  [3]Concordia University  [4]Mila  [5]Peking University
[*]Equal Contribution.  [‖]Part of the work was done during an internship at Meta.
Correspondence to `yf2231@nyu.edu`

## Abstract

Large Language Models (LLM) are increasingly trained on data generated by other LLMs, either because generated text and images become part of the pre-training corpus, or because synthetized data is used as a replacement for expensive human-annotation. This raises concerns about *model collapse*, a drop in model performance when their training sets include generated data. Considering that it is easier for both humans and machines to tell between good and bad examples than to generate high-quality samples, we investigate the use of verification on synthesized data to prevent model collapse. We provide a theoretical characterization using Gaussian mixtures, linear classifiers, and linear verifiers to derive conditions with measurable proxies to assess whether the verifier can effectively select synthesized data that leads to optimal performance. We experiment with practical tasks – computing matrix eigenvalues with transformers and news summarization with LLMs – which both exhibit model collapse when trained on generated data, and show that verifiers, even imperfect ones, can indeed be harnessed to prevent model collapse and that our proposed proxy measure strongly correlates with performance.

## 1 Introduction

As generative models for language, images, and video continue to achieve human-level performance (Touvron et al., 2023; Achiam et al., 2023; Ramesh et al., 2021; Rombach et al., 2022; OpenAI, 2024), they are increasingly used to synthesize data across diverse domains, including coding (Haluptzok et al., 2022) and mathematics (Trinh et al., 2024). With this rise in AI-generated data, there is growing interest in its potential to replace expensive human annotators.

This gradual replacement of human-written corpora by machine-generated tokens gives rise to a number of concerns, notably the risk of "model collapse" (Shumailov et al., 2023; 2024), where iterated training on synthesized data brings a drop in model performance, and, ultimately, "dumber models". The phenomenon was observed empirically in (Hataya et al., 2023; Martínez et al., 2023a;b; Bohacek & Farid, 2023; Briesch et al., 2023; Guo et al., 2023; Taori & Hashimoto, 2023) and theoretically analyzed in (Alemohammad et al., 2023; Bertrand et al., 2023; Dohmatob et al., 2024c;a). In particular, Dohmatob et al. (2024c;a) show that model collapse is an effective change of scaling law: as data becomes more synthetic, larger training sets do not enhance performance. Current thinking around model collapse recommends that synthetic data is to be avoided for model training lest the system deteriorate completely.

This raises a critical question: are synthesized data so fundamentally flawed that they must be identified and discarded using detection techniques (Zhang et al., 2024b), corrected through various refinement methods (Gillman et al., 2024) or used as negative feedback (Alemohammad et al., 2024)? In this work, we argue that synthesized data contain abundant valuable information that can be effectively leveraged. We advocate for a paradigm of verifier-based selection of synthesized data and provide criteria for when this process is successful - even with imperfect verifiers.

Throughout this paper, we focus on a scenario where synthesized data serve as the labels (answers) to questions—a common framework in tasks such as question answering, code generation, and mathematical reasoning. To begin, we explore the information contained in synthesized data through a controlled experiment: training transformers on a linear algebra task. This setting allows us to

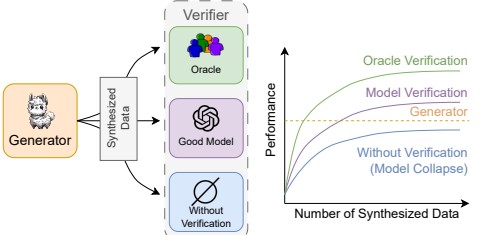 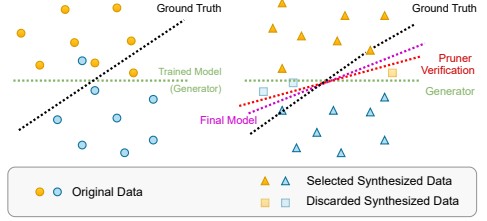

(a) The generator is capable of producing high-quality data, and verifiers can extract this data through selection to prevent model collapse, which occurs in the absence of such selection.

(b) In theory, we consider a Gaussian mixture model with a linear generator and linear pruner. The pruner improves synthesized data through selection, resulting in improved performance.

Figure 1: Illustrative figures for our proposal (**a**) and for the theoretical and simulation settings (**b**).

both understand the solutions and easily synthesize data, placing it into a long established corpus of works that study interesting phenomena in a controlled setting to advance our understanding of the underlying mechanisms in larger models in the wild. For example, see (Power et al., 2022; Garg et al., 2022; Charton, 2024; Dohmatob et al., 2024c). In this context, we find that the accuracy of the most accurate synthesized solution among the top candidates is three times higher than that of the solution selected by the model itself based on perplexity — its own measure of quality. This finding provides a positive answer to our question: the model is indeed capable of generating high-quality solutions. However, it cannot intrinsically *identify* the best solution using perplexity alone. Thus, effectively scaling with synthesized data requires a robust selection and verification process.

We proceed to provide a theoretical characterization using Gaussian mixtures and linear classifiers, employing an external verifier to select the generated data. In the limit of infinite synthesized data, we analyze the conditions required for the generator and verifier to enable the model, trained on selected synthesized data, to achieve Bayes-optimal results. Notably, there is a sharp phase transition — from zero accuracy due to errors in the synthesized data to optimal accuracy. Our theory confirms that there is abundant information within the synthesized data, and that proper, not necessarily perfect, selection can overcome model collapse. We also identify a measurable proxy function that characterizes the ability of the verifier to select synthesized data that results in good models when retraining on it.

We further conduct large-scale experiments to test our theoretical insights : (1) the above mentioned transformers on a linear algebra task and (2) news summarization using the LLM Llama-2 (Touvron et al., 2023). In both experiments, we apply various verifiers for selection and compute our theoretical proxy function. Our results show that relying solely on generated data leads to poorer performance than that of the generator itself, even with increased data volume, indicating model collapse. However, even with noisy and imperfect selection methods, synthesized data can be improved to yield models that outperform the original generator. Our proxy function strongly correlates with final performance across all cases. In particular it elucidates the counter-intuitive fact that a stronger model is not automatically a better selector — as we demonstrate by showing inferior performance when selecting with Llama-3 compared to self-selection with Llama-2 for Llama-2-generated text.

We summarize our contributions as follows:

- We provide a theoretical analysis in the high-dimensional limit with infinite data proving that synthesized data with appropriate selection can lead to optimal performance, and provide a characterization of necessary conditions via a measurable proxy function (Section 4).
- We validate our theoretical findings in three empirical settings of increasing departure from our theoretical assumptions:
    - Simulation results with linear classifiers, tested in a finite-data regime (Section 5),
    - Linear algebra tasks with transformers trained in a generative way (Section 6.1),
    - News summarization using Llama-2 (Section 6.3).

## 2 RELATED WORK

We here only list works of direct relevance to ours, with an extensive reference list in Appendix A.

**Model Collapse.** With the advancement of generative models, synthesized data generated by these models has become increasingly prevalent online, mixing irreversibly into our training corpora. Recent studies have highlighted the potential for dramatic deterioration in downstream models, a

phenomenon known as *"model collapse"* (Shumailov et al., 2023). Empirical studies have demonstrated this issue in various settings (Hataya et al., 2023; Martínez et al., 2023a;b; Bohacek & Farid, 2023; Briesch et al., 2023; Dohmatob et al., 2024b; Yang et al., 2025; Askari-Hemmat et al., 2025). Synthetic data can also amplify biases through feedback loops (Taori & Hashimoto, 2023). Synthesized datasets have been shown to reduce diversity (Padmakumar & He, 2024; Guo et al., 2023) and cause distributional distortions (LeBrun et al., 2021). Theoretical analysis also examines the effects of iterative training on self-generated data (Alemohammad et al., 2023; Bertrand et al., 2023; Dohmatob et al., 2024a; Seddik et al., 2024). Notably, Dohmatob et al. (2024c) warn that model collapse signifies a break in customary neural scaling laws (Kaplan et al., 2020; Hoffmann et al., 2022), where increasing synthesized data volume does not enhance performance as effectively as scaling with human-generated data. As a result, recent works have focused on avoiding or correcting synthetic data to prevent model collapse. Gillman et al. (2024) propose using a correction function informed by expert knowledge to modify the synthesized data. Alemohammad et al. (2024) leverage a model trained on synthetic data as negative guidance for diffusion models. Zhang et al. (2024b) employ the confidence score and an AI detection classifier to discard synthesized data. In contrast, we propose leveraging the synthesized data through selection techniques.

**Benefits of Synthesized Data.** Synthetic data holds great potential, as it is much easier and cheaper to scale compared to human-labeled data. Numerous empirical studies have demonstrated the benefits of synthesized data across a wide range of settings. Common practices include cases where the downstream task slightly differs from that of the data-generating model, where the generating model is significantly stronger than the consuming one, or when better prompt engineering and external information are utilized. In Appendix A.1, we provide a taxonomy that outlines when and how synthesized data can be advantageous. Data selection is already employed in some works, particularly in code generation and mathematics, where natural verifiers such as compilers, solutions, or heuristic verifiers exist. For instance, Haluptzok et al. (2022) generate synthesized code and filter out incorrect samples. Ulmer et al. (2024) use conversational metrics to filter synthetic dialogue data. Trinh et al. (2024) utilize a symbolic deduction engine to verify correct solutions for Olympiad geometry problems. Setlur et al. (2024) apply a final answer verifier to distinguish between good and bad synthetic data. Although verifiers are used in these cases, their effects on performance have not been systematically explored, especially in terms of how different types of verifiers influence outcomes.

**Data Selection.** Data selection is a preprocessing technique typically applied to original datasets with human labels, as discussed in various related works (see Appendix A.3). In contrast, our work focuses on data selection for *synthesized* data, which does not represent a noisy version of the ground truth but rather a skewed distribution. Our aim is not to propose new selection methods but to demonstrate how selection can prevent model collapse and to analyze the conditions under which it is effective.

## 3 WARMUP

We first focus on transformer models on a mathematical task, which offers an interpretable setting to understand the generation quality since we have a clear metric for error and a computable ground truth, as well as the ability to generate as much (original) data as desirable.

**Setting.** We follow Charton (2022), who showed that transformers (Vaswani et al., 2017) can learn to predict the eigenvalues of $5 \times 5$ real symmetric matrices with independent and identically distributed entries, rounded to three significant digits. All training, test, and synthesized data are generated by sampling matrices with independent entries from $\mathcal{U}[-10, 10]$. A prediction is considered correct if the relative error in the $L^1$ norm is below a certain tolerance $\tau$. The synthesized data generator is trained on a limited sample of 200,000 examples with Adam for 65 epochs. Details on the tokenizer and optimization can be found in Appendix B.

Table 1: **Generator accuracy for different beam sizes.** Left: the most accurate solution in the beam is evaluated. Right: the solution with lowest perplexity is evaluated.

| Tolerance $\tau$ | Verify all beams | | | Verify the best beam | | |
|---|---|---|---|---|---|---|
| | 2% | 1% | 0.5% | 2% | 1% | 0.5% |
| Beam 50 | **90.4** | **60.4** | **22.9** | **65.9** | **19.2** | **2.4** |
| Beam 25 | 88.0 | 53.2 | 16.8 | 66.1 | 19.3 | 2.4 |
| Beam 10 | 83.7 | 43.1 | 10.5 | 66.2 | 19.5 | 2.5 |
| Beam 5 | 79.3 | 34.9 | 7.1 | 66.5 | 19.7 | 2.4 |
| Greedy | **66.9** | **20.2** | **2.4** | **66.9** | **20.2** | **2.4** |

We aim to evaluate the quality of the generator's outputs and determine how much useful information can be extracted from it. To achieve this, we employ *beam search*, an inference technique that

maintains multiple candidate sequences (referred to as "beams") during the generation process to explore a larger portion of the solution space. Beam search tracks the top $k$ candidate sequences based on their cumulative probability scores, expanding each by considering all possible next tokens. When $k = 1$, beam search is equivalent to greedy decoding. The algorithm continues this expansion until the full generation process is complete, allowing us to assess the upper bound of the generator's capabilities and better understand its limitations and strengths.

In Table 1, we report the test accuracy of generated predictions using beam search. We consider two settings: evaluating only the best beam identified by the model (right) to understand the model's actual predictions, and another evaluating all $k$ candidates in the beams, with the best one contributing to the accuracy (left) to assess the model's full potential.

For self-selection, increasing the number of beams does not lead to any improvement in accuracy, despite significantly higher inference costs. However, in the left setting, moving from greedy decoding (beam 1) to beam 50 improves accuracy from 20.2% to 60.4%, with $\tau = 1\%$. This indicates that **the model has considerable potential to generate improved solutions, but it lacks the inherent capability to autonomously select the best predictions**. Among the vast amount of synthesized data, there may be high-quality data, but effective selection is necessary.

## 4 THEORETICAL INSIGHTS

In this section, we aim to theoretically explore training and verification using synthesized data, and characterize the conditions under which verification enhances performance. To streamline our analysis, we focus exclusively on training with synthesized data, since the inclusion of additional real data is always advantageous. We consider learning with a family of high-dimensional data distributions. Within this context, the verification process is implemented as a *pruning strategy* applied to synthesized data. Crucially, we will not assume that the pruning strategy has access to the ground truth, as such an assumption would be overly restrictive for practical applications; rather, we will formulate our theory for general pruners, which could for instance be like another trained model. A full exposition of our general theory is in Appendix E; for ease of exposition, we specialize here to Gaussian mixtures.

### 4.1 SETTING

**Data Distribution.** We will consider distributions $P$ over $\mathbb{R}^d \times \{0, 1\}$ with certain high dimensional concentration properties of a general form (Condition E.1). A special case are binary *Gaussian mixtures*: features have conditional distribution given by $x \mid y \sim N(\mu_y, \Sigma)$, where $\mu_y = (2y - 1)\mu$, for some $\mu \in \mathbb{R}^d$ and $\Sigma$ is a positive-definite matrix with $\mathbb{E} \|x\|^2 = \|\mu\|_2^2 + \operatorname{tr} \Sigma = 1$. For further ease of exposition we will only consider balanced distributions:

$$\mathbb{P}(y = 1) = \mathbb{P}(y = 0) = 1/2, \text{ for } (x, y) \sim P.$$

**Synthesized Data.** Let $D_N = \{(x_1, y_1), \ldots, (x_N, y_N)\}$ be a dataset of $N$ i.i.d. pairs from the true distribution $P$ and let $D'_N = \{(x_1, y'_1), \ldots, (x_N, y'_N)\}$ be the synthesized data generated from the same distribution, but where label $y'_i$ (instead of $y_i$) has been generated by an AI model. Further, we shall denote by $p \in [0, 1)$, the probability that the label $y'_i$ of a synthesized example $(x, y'_i)$ is different from the true label $y_i$, i.e

$$p := \mathbb{P}(y'_i \neq y_i). \tag{1}$$

Note that $p$ does not depend on the example index $i$, due to the i.i.d. assumption. We make no further assumptions on the synthesize data.

**Downstream Model and Pruning.** We will model our data selection (whether with or without feedback) via a *pruning strategy* $q = (q_1, \ldots, q_N)$ where $q_i$ is a bit which indicates whether the $i$th training example from $D'_N$ has survived pruning. For the downstream models we consider the family:

$$\mathbb{P}(y = 1 \mid x, w) = \hat{y} := \sigma(x^\top w) \in (0, 1), \qquad \sigma(z) := \frac{1}{1 + e^{-z}}$$

parametrized by a vector of weights $w \in \mathbb{R}^d$ and sigmoid non-linearity $\sigma$. Let $\widehat{w}_N$ be obtained via logistic regression fitted on $D'_N$ with ridge regularization parameter $\lambda > 0$. Thus, $\hat{w}$ is the unique

minimizer of the following objective function:

$$L(w) := \frac{1}{N} \sum_{i=1}^{N} q_i \ell(\sigma(x_i^\top w), y_i') + \frac{\lambda}{2} \|w\|^2, \tag{2}$$

where $\ell$ is the binary cross-entropy. The corresponding downstream classifier is $\widehat{f}_N = f_{\widehat{w}_N}$, where the notation $f_w$ refers to the linear classifier induced by a weights vector $w \in \mathbb{R}^d$, i.e $f_w(x) = (\text{sign}(x^\top w) + 1)/2$. **Test Accuracy.** The test accuracy of the downstream model $\widehat{f}_N$ is defined by

$$acc(\widehat{f}_N) := \mathbb{P}(\widehat{f}_N(x) = f_{\text{Bayes}}(x)), \text{ for a random test point } (x, y) \sim P,$$

where $f_{\text{Bayes}}(z) := \mathbb{E}[y|x = z]$ is the Bayes-optimal classifier. In particular, note that $acc(f_{\text{Bayes}}) = 100\%$ by construction. The quantity $acc(\widehat{f}_N)$ will be the main object of our analysis, and we will be interested in how it depends on errors in the generator $P$ and the choice of pruning strategy $q$, in the infinite-sample limit $N \to \infty$.

## 4.2 PRUNING STRATEGY

We consider a wide class of parametrized pruning strategies $q$, which we term *Verification-Pruning* (see Appendix E). They satisfy the following reasonable property:

**Assumption 4.1** (Independent Selection)**.** *The bits $q_1, \ldots, q_N \in \{0, 1\}$ are independent. Thus, in particular, whether any training example $e_i := (x_i, y_i') \in D_N'$ survives pruning or not is independent of what happens to the other examples $e_{j \neq i}$.*

In its most general form, our verification-pruning family (refer to Appendix E for details) is described by four parameters $(\phi_0, \phi_1, \psi_{01}, \psi_{10})$, defined as follows:

$$\phi_k = \mathbb{P}(q = 1 \mid y' = k, y = k), \quad \psi_{k\ell} = \mathbb{P}(q = 1 \mid y' = \ell, y = k). \tag{3}$$

Assumption 4.1 implies that for any class labels $k, \ell \in \{0, 1\}$, the random variables $(z_{ik\ell})_{i \in [N]}$ defined by $z_{ik\ell} = 1[y_i = k, y_i' = \ell, q_i = 1]$ are i.i.d. with Bernoulli($p_{k\ell}$) distribution, with

$$p_{kk} = (1 - p)\phi_k/2, \text{ and } p_{k\ell} = p\psi_{k\ell}/2 \text{ if } k \neq \ell, \text{ with } p \in [0, 1) \text{ as in equation 1.} \tag{4}$$

For simplicity of exposition, we will focus on *symmetric* pruning strategies where $\phi_1 = \phi_0 = \phi$ and $\psi_{01} = \psi_{10} = \psi$. Such pruning strategies are therefore described by a pair of parameters $(\phi, \psi)$.

Let us provide two more notable examples of (symmetric) pruning strategies.

**No Pruning.** The case $(\phi, \psi) = (1, 1)$ corresponds to no pruning, i.e using the entire training dataset.

**Oracle Pruning.** The case $(\phi, \psi) = (1, 0)$. The pruning strategy only keeps indices corresponding to examples in the dataset which have correct label (all corrupted labels are discarded).

**Supervised Pruning.** We shall also consider *supervised* pruning strategies $q$ defined by

$$q_i = 1[y_i'(x_i^\top w_{prune}) > 0], \tag{5}$$

for some $w_{prune} \in \mathbb{R}^d$. This pruning strategy filters out all examples on which there is disagreement on the assigned label. In Appendix E we show how we can map this to $(\phi, \psi)$-pruning.

## 4.3 PERFORMANCE OF MODELS TRAINED WITH PRUNING: INSIGHTS FROM INFINITE-SAMPLE REGIME

The following is our main theoretical result (see Theorem E.3 for full statement). It characterizes test accuracy $acc(\widehat{f}_N)$ of the downstream model on pruned data as a function of $p$ (the label disagreement) and the parameters $(\phi, \psi)$ of the pruner, in the theoretical limit of infinite training data ($N \to \infty$).

**Theorem 4.2** (Simplified version of Theorem E.3)**.** *Let Assumption 4.1 be in order. Fix $p$, $\phi$, $\psi$ and define the breakdown point $p_\star \in (0, 1)$ by $p_\star := 1/(1 + \psi/\phi)$. For the family of data distributions obeying Condition E.1 (including the Gaussian mixture), for a downstream model $\widehat{f}_N$ trained on data from a generator with error rate $p$, pruned with an verification-type strategy with parameters $(\phi, \psi)$, in the limit $N \to \infty$ it holds a.s that: (i) If $p < p_\star$ then $acc(\widehat{f}_N) = 100\%$. (ii) If $p > p_\star$ then $acc(\widehat{f}_N) = 0\%$. The pruner is overwhelmed by so many inaccuracies in the synthesized data, and the downstream model learns the exact opposite of the true class labels.*

Thus, there is a sharp phase-transition around the corruption level $p_\star := 1/(1 + \psi/\phi)$: as $p$ is increased past level $p_\star$, the downstream model $\widehat{f}_N$ abruptly switches from being perfectly accurate, to perfectly inaccurate! The proof (see Appendix E.7 for a sketch) explicitly computes empirical test accuracy in terms of $N_{k\ell} := \sum_{i=1}^{N} z_{ik\ell}$, which follow a binomial distribution, bounding the gap to the population accuracy, and using concentration of measure type techniques. Note that the sharp transition is due to the infinite-sample regime, where we can avoid finite-sample corrections. $p_*$ could be used as a measurable proxy in the experiment.

See Figure 2 (and Figure 6 in Appendix E) for an empirical illustration of the theorem.

**Remark 4.3.** *Note that the* $100\%$ *accuracy achievable in Theorem 4.2 is idealized, and is expected to only hold in the infinite sample regime (with a possibly large but fixed input dimension).*

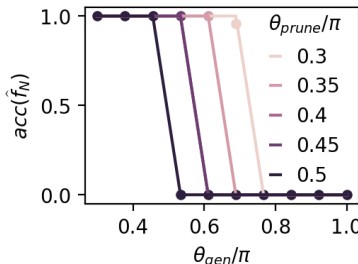

Figure 2: **Empirical confirmation of Theorem 4.2.** Comparing the breakdown points of different generators and pruners of different strengths. Synthesized data is generated from a linear model $w_{gen}$ with classification error rate $p = \theta_{gen}/\pi \in [0, 1]$. The data is pruned with another linear model $w_{prune}$ which has classification error $\theta_{prune}/\pi$. Broken lines correspond to the prediction of Theorem 4.2, while solid points correspond to experiments. Notice the sharp phase transitions where the model suddenly switches from perfect accuracy to worse-than-chance, as the theorem predicts.

## 4.4 SOME CONSEQUENCES OF THEOREM 4.2

We now present some illustrious applications of Theorem E.3. These examples are empirically confirmed in Figure 2 (see also Figure 6 in Appendix E).

**No Pruning.** Here, we have $\psi/\phi = 1$ and so the downstream model achieves $100\%$ accuracy for all values of corruption parameter $p$ up to the breakdown point $p_\star = 1/2$ predicted by Theorem E.3 .

**Oracle Pruning.** For this scenario, $\psi/\phi = 0$ and so Theorem E.3 predicts that the downstream model $\widehat{f}_N$ achieves $100\%$ accuracy for all values of corruption parameter $p$ up to the breakdown point $p_\star = 1$. This is perhaps not so surprising in hindsight. The point is that even for moderately large values of $\psi/\phi$, the breakdown point $p_\star$ can still be quite close to 1.

**Supervised Pruning.** Consider isotropic Gaussian mixture data with means $\pm\mu$, and a pruning strategy as in Eq. (5). The parameters $(\phi, \psi)$ only depend on the angles $\theta_{gen}, \theta_{prune}, \theta \in [0, \pi]$:

$$
\begin{aligned}
\theta_{gen} &:= \angle(w_{gen}, \mu), \ \theta_{prune} := \angle(w_{prune}, \mu), \\
\theta &:= \angle(w_{prune}, w_{gen}).
\end{aligned}
\tag{6}
$$

This is because, the $p_{k\ell}$'s defined in equation 4 now correspond to *orthant probabilities* for a trivariate normal distribution, with correlation coefficients given by these angles (see also Figure 2). In the Appendix E.6, we provide the calculation from the angles to $\psi$ and $\phi$. In practice, we can directly measure the proxy $p_*$, which encapsulates all the aforementioned correlations.

**Decoupling the Generator and Verifier.** Although the generator and verifier are coupled together in supervised pruning, there are some intuitions that help us decouple them: (1) a better generator always improves performance, (2) when the verifier is poor, such as in cases of no pruning or random pruning, we have a low breakdown point and require a good generator to achieve good performance, and (3) when the verifier is sufficiently good, close to an oracle, the breakdown point is high, and any non-degenerate generator is sufficient.

## 5 SIMULATIONS ON SYNTHESIZED DATA

The theoretical results are based on the best-case scenario of having unlimited access to synthesized data and operating in the high-dimensional limit. In this framework, the impact of generator and verifier on performance is reflected in binary outcomes: $100\%$ or $0\%$. In this section, we present simulation results from finite regimes to illustrate the practical implications of our theoretical framework. Specifically, we explore how the generator and verifier influence performance, and how performance scales with an increasing, yet finite, volume of synthesized data.

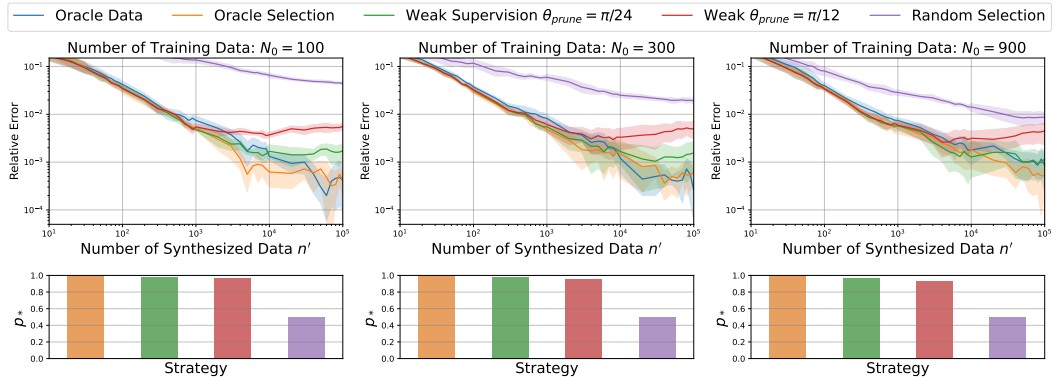

Figure 3: **Simulations with Gaussian mixtures. (Top row)** Relative error (accuracy relative to optimal accuracy) scaling as a function number of selected data, $n'$, used to train the model. $\tau = 0.15, N_1 = 10^6$. The Bayes optimal classifier achieves approximately 94% accuracy on this distribution. **(Bottom row)** $p_*$ values for all settings.

## 5.1 SETTING

Following the theoretical setting, we consider the same Gaussian mixture and linear models for the generator and selector. Let $w_*$ be a fixed unit vector in $\mathbb{R}^d$. The distribution $P_{orig}$ is

$$x|y \sim N(y\tau w_*, I_d/d), \quad \text{for } y \in [-1, +1].$$

Here, $\tau$ is a positive scalar that controls the overlap.

**Synthesized Data Generation and Verification.** We sample $N_0$ data points from the distribution $P_{\text{orig}}$ to form the original dataset, which is used to train a linear model, $\hat{w}_{N_0}$, employing ordinary least squares. We then generate a large synthesized dataset using the trained model $\hat{w}_{N_0}$ with a sigmoid function, which undergoes selection by various verifiers into sets $D_{\text{vrf}}^{\text{gen},N_0}$. In our study, these verifiers are linear models parameterized by various $w_{\theta_{prune}}$, where $\theta_{prune}$ denotes the angle between the pruner and the ground truth $w_*$. Having the verified synthesized data, we then randomly select $n'$ data from $D_{\text{vrf}}^{\text{gen},N_0}$ to train and evaluate the final models.

## 5.2 LESSONS LEARNED

In Figure 3, we conduct several simulations varying $N_0$, $\theta_{prune}$, and $n'$. A larger $N_0$ indicates a more effectively trained generator, utilizing a greater amount of original data. $\theta_{prune}$ determines the verifier's quality, with $\theta_{prune} = 0$ corresponding to the use of the ground truth as an oracle verifier. We further explore scenarios where $\theta_{prune} = \frac{\pi}{24}$ and $\frac{\pi}{12}$. The variable $n'$ represents the number of synthesized samples used to train the final model. We plot the scaling curves as $n'$ approaches infinity, a scenario analyzed by our theory. The "random" line represents the process of randomly selecting $n'$ data points from the synthesized dataset without any verification. The "clean" line indicates training of the model using $n'$ data points from the original distribution. We make the following observations:

**Oracle Verification Matches Training with Oracle Labels.** The oracle verifier achieves the best performance, matching training with clean data across all settings and attaining Bayes optimal accuracy. This validates the theory that synthesized data has the full information which can be extracted with verification.

**The Effectiveness of a Weak Verifier Depends on $n'$.** In practical scenarios, obtaining oracle-level verification is often challenging, so weaker forms of verification are typically employed. While a weak verifier generally leads to poorer performance, consistent with the decaying threshold points predicted by theory, there is a surprising "sweet spot" at a certain data size (for the curves with $\theta_{prune} = \frac{\pi}{12}$). Training around 10,000 data outperforms training with more data. Therefore, the choice of verifier should also take into account the quantity of data selected.

**$p_*$ is strongly correlated with the final performance.** Across all settings, the relative values of $p_*$ consistently align with the models' performance. The simulations provide practical validation of the theoretical results and are consistent with the interpretation presented in Section 4.4. Oracle supervision yields optimal performance, and, in general, improving both the generator and verifier leads to enhanced results.

Table 2: **Implementation of our three experiments**: We progressively explore our insights through three real-world experiments. First, we conduct simulations in a finite-data regime, where all settings align with theoretical expectations. Next, we examine transformers trained on generation tasks, evaluated using a 0-1 metric. Finally, we analyze large language models with general metrics.

| Settings | Data | Task | Model | Verifiers | Metric |
|---|---|---|---|---|---|
| Simulation | Gaussian mixture | classification | linear model | linear models | classification accuracy |
| Arithmetic | synthesized data | generation | transformer | (noisy) verifiers | accuracy w. tolerance |
| Summarization | XLSUM dataset | generation | Llama | Llamas | similarity w. Rouge-1 |

## 6 EXPERIMENTS

In both the theoretical results and the simulations, we examine a classification scenario where oracle supervision can select "100% correct" synthesized data. However, outside of these controlled settings, synthesized data may only approximate the ground truth, meaning the selected data might be closer to the truth rather than exactly correct. In this section, we explore verification for more general tasks using generative models through two experiments: (1) training a transformer to predict the eigenvalues of a matrix, and (2) fine-tuning Llama-2-7B on a news summarization task. Table 2 provides an overview of how we gradually relax the setting from theoretical to empirical. For the purposes of our discussion here, we refer to model collapse whenever the performance of the model trained on synthetic data is worse than that of the original generator.

### 6.1 TRANSFORMER FOR ARITHMETIC TASKS

Recall the setting described in Section 3, where a transformer is trained in a generative manner using 200K samples of matrices and their corresponding eigenvalues. This setting allows for the synthesis of unlimited data by generating random matrices and using the generator to predict their eigenvalues.

We first introduce the oracle **verifier** that measures the relative $L^1$ distance between the model's predictions and the correct solutions. We use greedy decoding to generate a prediction for each matrix, and only matrices with predictions within a tolerance of $\tau = 1\%$ (as determined by the oracle verifier) are retained. Additionally, we introduce a set of noisy verifiers, where data with predicted solutions exceeding the 1% tolerance are still included with a probability of $p_{\text{noise}}$ [1]. Since we are evaluating accuracy in a binary (0/1) manner for each data point, we can directly compute $\psi$ and $\phi$ in Equation 3, with $y' = y$ replaced by being within 1% relative $L^1$ tolerance. The noisy verifier corresponds to setting $\phi = 1$ and $\psi = p_{\text{noise}}$. Consequently, according to our theory, the proxy is given by $p_* = 1/(1 + p_{\text{noise}})$. For perfect oracle verifier, $p_{\text{noise}} = 0$ and $p_* = 1$; for random selection without verification, $p_{\text{noise}} = 1$ and $p_* = 0.5$.

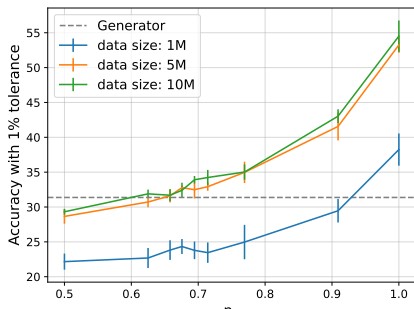

Figure 4: **Transformers computing eigenvalues.** Correlation between accuracy with 1% tolerance, $p_*$, and the number of synthesized data. Model collapse is observed without verification ($p_* = 0.5$), while higher values of $p_*$ result in improved performance. Results are averaged over 5 seeds.

We generate various verified synthesized datasets, with data sizes ranging from 1 million to 5 million, and up to 10 million samples. Using these datasets, the transformer is trained from scratch and evaluated with greedy decoding. We report how different verifiers, specifically different proxy values $p_*$, and varying dataset sizes affect the final performance in Figure 4. We observe the following:

**Model Collapse is Observed.** When using synthesized data without any verification, at $p_* = 0.5$, even with a significantly larger dataset — 10 million samples, or 50 times the size of the generator's original training set — the trained model performs worse than the generator itself (represented by the dashed line). This outcome indicates the occurrence of model collapse.

$p_*$ **as a Reliable Proxy for Final Performance.** When leveraging verification, we observe a consistent increase in performance. With the oracle verifier ($p_* = 1$), the accuracy nearly doubles. This observation aligns with practices in code generation and mathematics (Haluptzok et al., 2022;

---

[1]This is equivalent to introducing random noise in the selection of both correct and incorrect data.

Trinh et al., 2024), where a natural verifier, such as precomputed solutions, unit tests, or a compiler, is available. Across all dataset sizes, $p_*$ shows a strong correlation with the final performance. Furthermore, using 10 million samples approximates the effect of having infinite synthetic data; the turning point on the green curve, which surpasses the generator's performance, occurs around 0.65 in Figure 4. This value is close to the generator's error rate (i.e., 1 minus the accuracy, represented by the dashed line) and reflects the phase transition predicted by our theory.

## 6.2 GENERALIZATION OF THE PROXY

In the theory and the task of predicting eigenvalues, the evaluation is a two-way classification problem. For synthetic $y'$ and ground truth $y$, the indicator $s$, which measures the similarity, is a 0/1 variable. We can rewrite $p := 1 - \mathbb{E}[s]$ and define $\phi = \mathbb{P}(q = 1 \mid s = 1)$, $\psi = \mathbb{P}(q = 1 \mid s = 0)$. For general problems, we may instead have a distance measure $s$ between $y$ and $y'$. We extend the definition of $p_*$ under the assumption that $s$ is normalized to lie in $[0, 1]$, and redefine $\phi$ and $\psi$ as follows:

$$\phi = \mathbb{E}_s[qs]/(1 - p), \quad \psi = \mathbb{E}_s[q(1 - s)]/p.$$

When $s$ takes values in $\{0, 1\}$, these definitions are consistent with those in Equation (3). With this extension, we now move to consider general problems in NLP.

## 6.3 LLMS FOR NEWS SUMMARIZATION

We now turn to one of the most standard tasks in NLP: news summarization. We utilize the English summarization subset of the XLSUM dataset (Hasan et al., 2021), the largest publicly available summarization dataset, consisting of 307,000 training samples and 11,500 test samples. Each sample in this dataset contains a news article $x$ paired with a professionally annotated summary $y$. Unlike in the previous cases, we no longer have infinite $(x, y)$ pairs due to the finite number of news articles available. Therefore, we fine-tune a Llama-2-7B model (Touvron et al., 2023) on only $12.5\%$ of the training set to serve as the generator. The synthesized data (news summaries) is generated using the articles of the entire training set with greedy decoding. This approach reflects real-world conditions where synthetic data generation can significantly outpace human annotation. In this setting, we use the Rouge-1 score (Lin, 2004) to evaluate the generated summary $y'$. Rouge-1 assesses the quality of the generated summary by measuring the overlap of individual words between the generated and human-written summaries $y$.

We consider three selection strategies: (1) **Selection with Oracle**: We calculate the Rouge score between the generated summary and the ground truth summary, keeping the data with the highest scores; (2) **Selection with Llama-3**: We leverage a fine-tuned Llama-3 model with higher Rouge score than the generator and keep the data with the lowest perplexity; (3) **Self-Selection**: We use the generator to keep the data with the lowest perplexity. We apply three selection rates: $12.5\%$ (when the selected synthesized data is the same size as the original data used to train the generator), $25\%$, and $50\%$. For each combination, we collect the selected synthesized data to finetune the Llama-2 model. We present scaling law curves that illustrate how the Rouge-1 score improves with increasing amount of selected synthesized data used for training. Throughout the experiments, all finetuning was performed with full parameter training. Details are provided in Appendix D.

The results and $p_*$ are shown in Figure 5. We observe the following:

**Model Collapse is Observed.** The Random Selection curve represent training with synthesized data directly without verification. In Figure 5 **Left**, using the same amount of synthesized data results in worse performance compared to using the original data, indicating model collapse (comparing 'Random Selection' with 'Generator'). Only with more data, the Random Selection lines improve and nearly match the performance of the generator.

**Selection by Oracle Performs Best.** The dataset curated with oracle selection surpasses the performance of the generator in all settings. Oracle selection with $12.5\%$ of the data selected even surpasses the model trained with $100\%$ of the training set and original labels. The oracle verifier consistently has the best $p_*$ compared with the other methods.

$p_*$ **Strongly Correlates with Performance.** Surprisingly, self-selection leads to better performance than the generator, while Llama-3 verification results in performance similar to random selection. These findings seem counterintuitive, given that Llama-3 actually achieves a better Rouge-1 score.

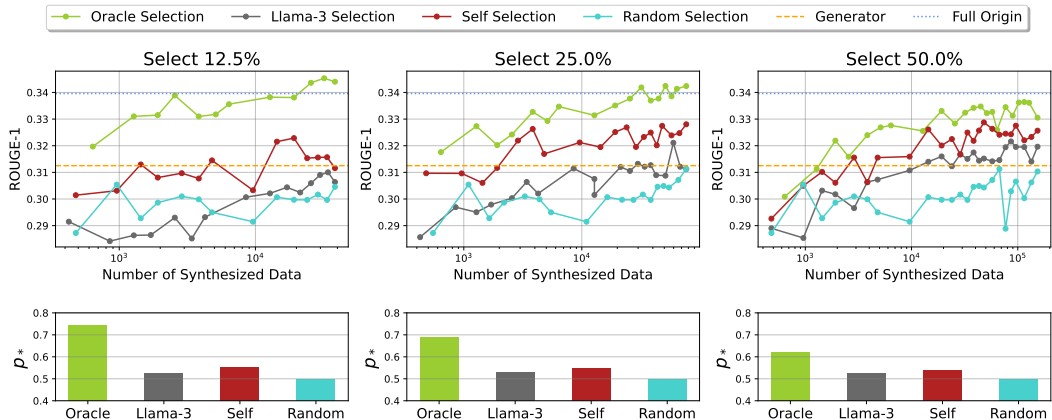

Figure 5: **News summarization with LLMs. (Top Row)** The three figures from left to right represents models trained on 12.5%, 25%, and 50% of the selected data individually. Each figure includes four curves illustrating different training scenarios: (1) selection with oracle, (2) selection with Llama-3 as a weak supervision, (3) self-selection, and (4) random selection. Additionally, two horizontal lines are included for comparison: one representing the generator model and the other representing a model trained with 100% data with original labels. **(Bottom row)** Computed values of $p_*$ for the verifiers for corresponding proportions of selected data. Training with data selected by a verifier with higher $p_*$ achieves better performance.

However, they are consistent with the measured $p_*$. Intuitively, self-selection tends to favor easy-to-learn samples, resulting in good performance with fewer data points. Although Llama-3 can generate better summaries, the synthesized data are produced by Llama-2, which is less correlated with Llama-3. As discussed in Equation 6, $p_*$ depends on three correlations and Llama-3 is not necessarily a better verifier in this context. This comparison highlights the challenges of selecting an appropriate verifier when a fixed oracle, like those used in code or math tasks, is not available. In such cases, our proposed $p_*$ can serve as a valuable proxy before any training is conducted. These observations resonate with the recent work (Fang et al., 2024) where it was also empirically observed that more accurate models might not always lead to better data filters.

# 7 DISCUSSION AND LIMITATIONS

In this paper, we consider a novel problem related to synthesized data: how to prevent model collapse through data selection. We propose to leverage a verifier to improve the synthesized data. We emphasize it is crucial to focus not only on the quality of the generator but also on having a high-quality verifier to select the data. We theoretically show that verification is all you need for synthesized data and identify a proxy function for performance after data selection. Through three solid experiments, we demonstrate that a decent selector can prevent model collapse and our proxy function is a reliable measure. Our work is of significant theoretical and practical importance in the era of large models with increasing use of generated data.

Can a model improve itself? In our paper, we present results showing that a math transformer using beam search does not improve test accuracy, while Llama-2, through self-selection of its generated data, can yield a model that performs better than the original generator. In the first experiment, the model selects a better $y$ (output) for each $x$ (input) (label selection). However, at the distribution level, the selected $y$ are not better. In the second experiment, the model selects better $(x, y)$ pairs from a larger pool of news articles ($x$) than available in the generator's training set. This introduces new information through the $x$ and results in a shift in the distribution of $x$. Due to this distribution shift, the trained model can outperform the original generator even with self-selection.

One limitation of this study is that we only considered data selection as a means of data curation. Besides data selection, data curation also includes methods such as data augmentation, data regeneration, and weighting. The exploration of general data curation methods to avoid model collapse is left for future work.

## ACKNOWLEDGEMENTS

YF and JK acknowledge support through NSF NRT training grant award 1922658. Part of this work was done while JK and YF were hosted by the Centre Sciences de Donnees (CSD) at the École Normale Supérieure (ENS) in 2023/24, and JK and YF would like to thank CSD and ENS for their hospitality. YF and PY would like to thank Yanzhu Guo, Di He, Zhenyu He for discussions and suggestions. This work was supported in part through the NYU IT High Performance Computing resources, services, and staff expertise.

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

# A    MORE WORKS ON SYNTHESIZED DATA

## A.1    TAXONOMY FOR SYNTHESIZED DATA

Contrary to the phenomenon of model collapse, synthesized data has been shown to improve performance in numerous empirical studies. We now provide a taxonomy outlining when and how synthesized data is beneficial. Specifically, we identify four key components: *prompt engineering* ●, *knowledge from advanced models* ▲, *distributional shift and targeted curation* ■, and *external verifiers* ✦. Most empirical studies can be categorized based on one or more of these components. We use ●▲■and ✦to denote the components each reference leverages.

**Code and Math.** Haluptzok et al. (2022) ✦generate synthesized data for codes and use a verifier to filter and show that the model can "self-improve" with its own synthesized data. Gunasekar et al. (2023) ●■filter high-quality data from the web and prompt GPT-3.5 with a specially curated prompt set covering both quality and diversity. Wei et al. (2023) ●leverage a diverse and large set of open-source code snippets to curate code instructions as prompts with good coverage and high quality. Zheng et al. (2024); Trinh et al. (2024) ✦leverage a symbolic deduction engine as a verifier to test the correctness of each branch for solving Olympic geometry problems.

**Alignment.** During standard fine-tunings, synthesized data is often generated by a stronger model like GPT-4 Peng et al. (2023) ▲. Wang et al. (2023) ●✦use a good set of prompts and inputs with a heuristic verifier to filter out low-quality ones and maintain high diversity. Bai et al. (2022) ●■use the model itself to critique whether its own generation is harmful, given already harmful prompts with gold standards from humans. For alignment with reinforcement learning, (Ouyang et al., 2022) ●✦use humans as verifiers to compare synthesized data generated by the current model with a good set of prompts. Some papers propose reinforcement learning with AI feedback (RLAIF) (Lee et al., 2023) ●■that leverages another LLM as the verifier to match human verification. The verifier is a stronger model, instruct-tuned Palm2 L, while the network being trained is the Palm2 XS. However, (Yang et al., 2024) ●later found that using better prompts (self-improve) that direct harmful or harmless responses can surpass RLAIF. Yuan et al. (2024) ●achieve surprising results with iterative fine-tuning and generating good prompts with in-context learning. Recently, (Feng et al., 2025) discuss how to optimally sample synthetic data for alignment.

**Knowledge distillation.** Most papers in the knowledge distillation area involve using a better model to distill for general performance or specific tasks, with ●, ▲, and ■involved from case to case. One example is the tiny story cases (Eldan & Li, 2023) ●▲, where GPT-4 is prompted to generate stories for four-year-olds that are used to train GPT-Neo with good performance.

**Image Domain.** (Kirillov et al., 2023) and (Li et al., 2022) ■use a distributional shift from high-quality to low-quality data to label and curate a vast amount of unlabeled data. Specifically, (Li et al., 2022) also trains a verifier to filters high-quality data. Um et al. (2024) ▲■specifically curate minority groups with a diffusion model to enhance performance. He et al. (2023); Dunlap et al. (2023) ▲■generate synthesized data that aids in classification tasks by tailoring the synthesized data to match the model's learning objectives. Azizi et al. (2023); Hemmat et al. (2023) ▲■employ guided curation (with supervision) to curate data from diffusion models. Burg et al. (2023) find that while synthesized data from a diffusion model helps improve downstream tasks, such as classification, using the pre-training data of the diffusion model alone gives even stronger performance.

## A.2    KNOWLEDGE DISTILLATION WITH SOFT LABELS

Related to synthesized data, there is a long history of using synthesized labels in image classifications. In the domains of self-distillation and knowledge distillation (Hinton et al., 2015; Furlanello et al., 2018), data with soft labels generated from the teacher model can significantly improve the performance of the student model. These soft labels convey additional insights—referred to as 'dark knowledge'—that have been theoretically linked to specific advantageous adaptations. These include implicit biases that mitigate overfitting (Mobahi et al., 2020), mimicry of early stopping (Dong et al., 2019) for improved optimization under label noise (Das & Sanghavi, 2023), and adjustments to accommodate specific data structures (Allen-Zhu & Li, 2022). We only consider synthesized data with fixed labels as in the current practice of LLMs and diffusion models.

## A.3 Data Selection

Comprehensive surveys on data selection for language models can be found in Albalak et al. (2024), along with theoretical studies on selection in high-dimensional settings (Sorscher et al., 2022; Kolossov et al., 2024). Specifically, Kolossov et al. (2024) also explore the use of surrogate models for producing labels during selection, followed by curation of the original labels. In our study, selection is applied to synthesized data where original labels are not available, resulting in distinct phenomena compared to these approaches on original data.

# B  Predicting the Eigenvalues

We leverage the code base provided by Charton (2022) at `https://github.com/facebookresearch/LAWT` under the license CC BY-NC 4.0.

**Input and Tokenization.** Transformers are trained to predict the eigenvalues of $5 \times 5$ symmetric real matrices. Model inputs are sequences of 25 real entries, rounded to three significant digits, and tokenized as triplets of signs($+$ or $-$), mantissas (from `0` to `999`) and power of ten exponents (from `E-100` to `E100`). For instance, the $2 \times 2$ matrix,

$$\begin{pmatrix} 2.3 & 0.6035 \\ 0.6035 & -3.141 \end{pmatrix}$$

will be encoded as the sequence of 12 tokens: `+ 23 E-1 + 604 E-3 + 604 E-3 - 314 E-2`. Model outputs are vectors of 5 real eigenvalues, rounded to three significant digits, and tokenized as before, as triplets of sign, mantissa and exponent (the `P1000` encoding from Charton (2022)).

**Model and Optimization.** We train sequence-to-sequence transformers (Vaswani et al., 2017), with 4 layers in the encoder, and one in the decoder, 512 dimensions and 8 attention heads, to minimize a cross-entropy loss, using the Adam optimizer (Kingma & Ba, 2014), with a fixed learning rate of $5 \cdot 10^{-5}$, after an initial linear warm-up phase over the first 10,000 optimization steps. The model is trained for 400 epochs before overfitting.

**Evalution.** Model accuracies are measured on a held-out test set of examples not seen at training. Model predictions are evaluated by decoding the output sequence as a vector of 5 real numbers $(p_1, p_2, p_3, p_4, p_5)$, and assessing that a prediction $\mathbf{p}(p_1, p_2, p_3, p_4, p_5)$ of eigenvalues $\mathbf{v}(v_1, v_2, v_3, v_4, v_5)$ is correct if the relative error in $L^1$ norms is below some tolerance $\tau$, i.e. if

$$\sum_{i=1}^{5} |v_i - p_i| < \tau \sum_{i=1}^{5} |v_i|.$$

We use tolerances $\tau$ of $5, 2, 1$ and $0.5\%$.

## B.1 Finetuning Models with Synthesized Data

In all previous experiments, the data generated from the generator (using beam or reject sampling) were used to train a new model. In this section, we consider using data generated from the generator to finetune models pre-trained on a small sample of ground truth data. We consider four cases:

- Fine-tuning the generator (Model A).
- Fine-tuning a model pre-trained to the same accuracy as the generator (62%, Model B).
- Fine-tuning a model pre-trained to higher accuracy (93%, Model C).
- Fine-tuning a model pre-trained to low accuracy (4%, Model D).

Table 3 compares accuracy of the four fine-tuning cases to that of a model trained from scratch. Fine-tuning only achiueves better performance when the pre-teained model achieved higher accuracy than model A. In all other cases, fine-tuning brings no improvement. Note that fine-tuning model A on its own generated data achieves the worst result, a clear case of model collapse.

|            | Model A (66%) | Model B (62%) | Model C (93%) | Model D (4%) | From scratch |
|------------|:-------------:|:-------------:|:-------------:|:------------:|:------------:|
| Rejection  | 61.8          | 72.9          | 82.1          | 66.3         | 72.1         |
| Beam 50    | 74.1          | 82.6          | 87.3          | 78.3         | 84.0         |
| Beam 35    | 72.7          | 81.3          | 86.8          | 76.8         | 80.4         |
| Beam 25    | 71.3          | 79.8          | 84.4          | 73.3         | 79.9         |
| Beam 10    | 67.5          | 75.1          | 83.5          | 68.0         | 73.9         |
| Beam 5     | 64.9          | 70.8          | 80.1          | 65.6         | 69.1         |
| Beam 1     | 61.6          | 62.1          | 75.6          | 55.8         | 60.5         |

Table 3: **Performance of models fine-tuned on 1M examples generated by the generator.** $\tau = 2\%$

## B.2 COMPUTATIONAL RESOURCES

We leverage a V100 GPU with 32GB of memory for all experiments involving linear algebra. The training time ranges from 1 to 5 days, depending on the data size and the number of epochs.

## C GENERALIZATION OF THE PROXY

Recall that in our theoretical setting, we define all variables in a two-way classification problem, with $y$ as the ground truth label and $y'$ as the predicted synthesized label. We introduce an indicator variable $s$, defined as:

$$s = \begin{cases} 1 & \text{if } y = y' \\ 0 & \text{if } y \neq y' \end{cases}.$$

We can rewrite all formulas in terms of $s$:

$$p := 1 - \mathbb{E}[s],$$

and

$$\phi = \mathbb{P}(q = 1 \mid s = 1), \quad \psi = \mathbb{P}(q = 1 \mid s = 0).$$

When moving beyond the classification setting, $s$ is no longer a binary 0/1 variable. In more general cases, $s$ represents a measure of similarity or distance between the original target $y$ and the synthesized target $y'$. Without loss of generality, we assume $s$ is normalized to lie within the range $[0, 1]$. For instance, in the news summarization experiment, $s$ corresponds to the ROUGE-1 score between the ground truth text $y$ and the synthesized text $y'$, and equals 1 if and only if $y = y'$.

With this generalized measure $s$, we can extend the previous definitions of $\phi$ and $\psi$ as follows:

$$\phi = \frac{\mathbb{E}_s[qs]}{1 - p}, \quad \psi = \frac{\mathbb{E}_s[q(1 - s)]}{p}.$$

When $s$ takes values in $\{0, 1\}$, these definitions are consistent with those in Equation 3.

## D NEWS SUMMARIZATION

We leverage the XLSUM dataset (Hasan et al., 2021) at `https://huggingface.co/datasets/csebuetnlp/xlsum` under the license CC-BY-NC-SA 4.0.

**Data preprocessing.** For each data in both training and test dataset, it consists of a news report and a summarization, denoted as (*news*, *summarization*). We write each data in the following form:

> Article: *news*. A summary of the article: *summarization*.

**Fine-tuning and generating details.** Throughout all phases of evaluation and generation, we employ greedy decoding. Given that news summarization is a low-entropy task, greedy decoding is chosen to ensure quality generation. Consistent with common practice, fine-tuning is limited to a single epoch. Through out the experiments, all the finetuning is with full parameter tuning to better capture the scaling law as suggested in Zhang et al. (2024a).

**Implementation details.** We leverage the official implementation in Huggingface [2] for training, under the license Apache 2.0. Specifically, for training the generator, we start our training with the pre-trained Llama-2, and set the learning rate to 5e-5, the learning rate scheduler as 'cosine', the number of epochs to 1, the total batch size to 32, the block size to 1024 and the others to the default value. For generating the synthesized data, we use greedy strategy to generate a summarization for each news in the training set. For training based on the selected synthesized data, we also start our training with the pre-trained Llama-2, and set the learning rate to 2e-5, the learning rate scheduler as 'constant' and the others to the same. For evaluation, we first use greedy strategy to generate a summarization for each news in the test set, and then calculate the Rouge-1 score between the generated summarization and the corresponding ground truth, and finally report the average of the Rouge-1 scores of all test data. When calculating the perplexity, we only calculate the perplexity for the generated summary. When fine-tuning the Llama-3 model, we use the full XLSUM dataset to achieve good performance. The resulting model achieves a Rouge-1 score of 34.5.

**Computational Resources.** All experiments were conducted using a dedicated computational cluster equipped with 4 NVIDIA A800 GPUs, each with 80 GB of memory. Our training and inference processes are performed on the cluster.

**Estimated Time.** Training the whole dataset for an epoch takes about 6 hours. Generating the whole dataset takes about 1 day. During evaluation, we need to first generate and calculate the rouge score, which takes around 40 minutes for one checkpoint.

**Quantitative Results.** We show two additional qualitative examples from the testset of the news summarization task to compare how the model performs with each selection strategy on the same news as follow. Notice how the generation without selection collapses: it heavily relies on a clichéd format that lacks specific information. While this approach might work for simple news by piquing curiosity, it fails to convey the depth or significance of the current news. The model trained on generated samples collapses to these formats. The Llama-3 selection does not address this issue, aligning with our results that Llama-3 selection performs poorly despite using a better model. In contrast, generations with self-selection and oracle selection produce summaries that are specific, informative, and directly address the key event, significantly improving the quality of the output.

---

**Quantitative examples 1 in news summarization**

**Article**: Currently just the US dollar, the euro, the yen and the pound are in the group. The IMF said the yuan "met all existing criteria" and should become part of the basket in October 2016. IMF chief Christine Lagarde said it was "an important milestone in the integration of the Chinese economy into the global financial system". She added it was also a recognition of the progress that the Chinese authorities have made in the past years in reforming China's monetary and financial systems. The yuan will now make up part of the IMF's Special Drawing Rights (SDR) - an asset created by the IMF which serves almost as a currency. It is used for transactions between central banks and the IMF, and is used to decide the currency mix that countries like Greece, for example, receive when the IMF provides financial aid. The last change made to the basket was in 2000, when the euro replaced the German mark and the franc. China is the world's second largest economy behind the US and asked for the yuan to become a reserve currency last year. Analysis: Andrew Walker, BBC World Service economics correspondent More than anything this move is a symbol - a powerful one - of China's meteoric rise, from poverty to pillar of the global economy. Until now only the currencies of the four leading developed economies were used in the basket that determines the value of the SDR. So for China to join this very small club is quite a statement of how the world economy has changed. There could also be some real benefits to China. If the currency's elevated status leads to more being held by central banks businesses and businesses, it would be helpful for the government's finances. A state's monopoly over the issue of currency indirectly gives it an extra source of revenue. But the main point is that the decision is new way of conveying the increasingly apparent message that China has a central role in driving the performance of the world economy. Vaswani: The Chinese yuan is going global What's next? Some analysts have suggested that by 2030 the yuan will become one of

---

[2]https://github.com/huggingface/transformers/blob/main/examples/pytorch/language-modeling/run_clm.py

the top three major international currencies, together with the dollar and the euro. Concerns about Beijing keeping the yuan artificially low to help exporters is one reason the currency has previously failed to meet the criteria for reserve currencies set out by the IMF. However, Chinese officials have a made a concerted effort to build support for the yuan's inclusion, and a recent IMF staff report endorsed such a move. Initially, the currency's inclusion would be largely a symbolic gesture, some analysts have said. They have also said the yuan's continuing inclusion in the basket would depend on whether China progresses with its financial reforms.
**Generation of the model trained with oracle selection**: China's yuan has been added to the International Monetary Fund's basket of reserve currencies, the IMF has announced.
**Generation of the model trained with Llama-3 selection**: 2015 was a good year for China's currency.
**Generation of the model trained with Self-selection**: The International Monetary Fund (IMF) has approved the inclusion of the Chinese yuan in its basket of reserve currencies.
**Generation of the model trained without selection**: 2015 will be a big year for the Chinese yuan.
**Ground truth**: The International Monetary Fund (IMF) has announced that China's currency, the yuan, will join the fund's basket of reserve currencies.

---

Quantitative examples 2 in news summarization

**Article**: His brother Mahinda Rajapaksa is expected to be installed as prime minister, having held the role as caretaker since November. The brothers' party, Sri Lanka People's Front, has secured a two-thirds "super majority" of seats needed to carry out its promised constitutional changes. The party won 145 of the 225 seats, plus five more seats from its allies. Mahinda Rajapaksa earlier tweeted that Indian Prime Minister Narendra Modi had called to congratulate him. The controversial Rajapaksa family has dominated Sri Lankan politics for two decades. Mahinda Rajapaksa was previously president, from 2005 to 2015. The opposition of former prime minister Ranil Wickremesinghe has been trounced, having lost all but one of the 106 seats it held in the outgoing parliament. The main opposition party is now a new group established by the son of Ranasinghe Premadasa, a former president who was assassinated in 1993. Sri Lanka has been one of the few nations to hold an election despite the coronavirus pandemic. The vote had already been postponed twice because of the virus. The country has had relatively few confirmed infections and deaths of coronavirus - with a total of 2,839 cases and 11 deaths. Another victory for controversial brothers By Anbarasan Ethirajan, BBC South Asia Analyst Once again it's a huge victory for the Rajapaksa brothers. Just nine months after his impressive win in the presidential election, Gotabaya Rajapaksa has led his Sri Lanka People's Front to a two-thirds majority. He is hugely popular among the Sinhala majority for crushing the Tamil Tiger separatist rebels in 2009 when he was defence secretary. Many in the country also credit his administration for bringing stability and successfully containing the coronavirus outbreak. Gotabaya Rajapaksa is accused of human rights abuses during the civil war and also of targeting those who dissented. He always dismissed such accusations - but they have not gone away. A surge in Sinhala nationalism in the run-up to the election has also worried Sri Lanka's minority communities. Muslim leaders say their community is still reeling from the vilification that followed the devastating Easter Sunday suicide attacks by Islamist militants last year, which killed more than 260 people. With their dominant majority, the Rajapaksas could attempt to change the constitution, increase the powers of the president so reversing the work of the previous government to introduce more checks and balances. Activists, already alarmed by the diminishing space for dissent and criticism, fear such an eventuality could lead to ever greater authoritarianism.
**Generation of the model trained with oracle selection**: 70-year-old Gotabaya Rajapaksa has won a landslide victory in Sri Lanka's parliamentary election.
**Generation of the model trained with Llama-3 selection**: 72-year-old Gotabaya Rajapaksa has won Sri Lanka's presidential election, securing a two-thirds majority in parliament.
**Generation of the model trained with Self-selection**: 70-year-old Gotabaya Rajapaksa has won a landslide victory in Sri Lanka's parliamentary election.
**Generation of the model trained without selection**: 145 seats needed for a two-thirds majority 106 seats won by the main opposition party 49 seats won by the ruling party 13 seats

> won by the Tamil National Alliance 5 seats won by the Janatha Vimukthi Peramuna 1 seat won by the United National Party 1 seat won by the Jathika Hela Urumaya 1 seat won by the Civic Force
>
> **Ground truth**: Sri Lankan President Gotabaya Rajapaksa has declared victory in the country's parliamentary election.

# E  A GENERAL THEORY OF PRUNING WITH VERIFICATION

In Section 4 we have presented a special case of our general theory, which we describe here in more generality and detail. While some of our exposition here overlaps with Section 4, we prefer to leave it as a complete text that provides a stand-alone overview.

## E.1  DATA DISTRIBUTION

Consider a probability distribution $P$ over $\mathbb{R}^d \times \{0, 1\}$ with the following high-dimensional property

**Condition E.1.** *Given $N \le N(d)$ i.i.d. samples $(x_1, y_1), \dots, (x_N, y_N)$ from $P$ with $N \le N(d)$, the following hold estimates w.p $1 - o(1)$ uniformly on all $i, j \in [N]$, in the limit $d \to \infty$*

$$\|x_i\|^2 \simeq 1,$$

$$x_i^\top x_j \simeq \begin{cases} a, & \textit{if } y_i = y_j, \\ b, & \textit{if } y_i \ne y_j \end{cases}$$

*where $b < a < 1$ are constants. For simplicity of presentation of our results, We will further assume that $b = -a$ or $b = 0$.*

The above structural condition is inspired by an assumption in Das & Sanghavi (2023).

For simplicity of exposition, we will only consider balanced distributions, meaning that

$$\mathbb{P}(y = 1) = \mathbb{P}(y = 0) = 1/2, \text{ for } (x, y) \sim P.$$

**Gaussian Mixture Example.**  As a first example, in the case of Gaussian mixtures where the features have conditional distribution given by

$$x \mid y \sim N(\mu_y, \Sigma), \tag{7}$$

$$\tag{8}$$

where $\mu_y = (2y - 1)\mu$, for some $\mu \in \mathbb{R}^d$ and positive-definite matrix $\Sigma$ with $\mathbb{E}\|x\|^2 = \|\mu\|^2 + \operatorname{tr}\Sigma = 1$, we may take

$$a = \|\mu\|^2, \quad b = -a. \tag{9}$$

Condition E.1 then holds thanks to concentration, with $N(d) = e^{\Theta(d)}$.

## E.2  TRAINING DATA, DATA PRUNING, AND DOWNSTREAM MODEL

Let $D_N = \{(x_1, y_1), \dots, (x_N, y_N)\}$ be a dataset of $N$ i.i.d. pairs from the true distribution $P$ and let $D'_N = \{(x_1, y'_1), \dots, (x_N, y'_N)\}$ a version of the dataset (also i.i.d.) with labels $y'_i$ instead of $y_i$. For example, this could be labels generated by an AI trying to reproduce real-world data. $D'_N$ is the data on which the downstream model is trained.

We will consider a family of models given by

$$\mathbb{P}(y = 1 \mid x, w) = \hat{y} := \sigma(x^\top w) \in (0, 1),$$

parametrized by a vector of weights $w \in \mathbb{R}^d$. Here, $\sigma$ is the sigmoid function defined by

$$\sigma(z) := \frac{1}{1 + e^{-z}}. \tag{10}$$

For the loss function, we use binary cross-entropy (BCE), defined by

$$\ell(\hat{y}, y) = -y \log \hat{y} - (1 - y) \log(1 - \hat{y}). \tag{11}$$

Let $\widehat{w}_N$ be obtained via logistic regression fitted on $D'_N$ with ridge regularization parameter $\lambda > 0$. Thus, $\hat{w}$ is the unique[3] minimizer of the following objective function:

$$L(w) := \frac{1}{N} \sum_{i=1}^{N} q_i \ell(\sigma(x_i^\top w), y'_i) + \frac{\lambda}{2} \|w\|^2.$$

Here $q_i$ is a bit which indicates whether the $i$th training example has survived pruning. The numbers $q = (q_1, \dots, q_N)$ is called a *pruning strategy*. The corresponding downstream classifier is $\widehat{f}_N = f_{\widehat{w}_N}$, where the notation $f_w$ refers to the linear classifier induced by a weights vector $w \in \mathbb{R}^d$, i.e

$$f_w(x) := \begin{cases} 1, & \text{if } x^\top w > 0, \\ 0, & \text{otherwise.} \end{cases} \tag{12}$$

The test accuracy of the downstream model $\widehat{f}_N$ is defined by

$$acc(\widehat{f}_N) := \mathbb{P}(\widehat{f}_N(x) = f_{Bayes}(x)), \text{ for a random test point } (x, y) \sim P,$$

where $f_{Bayes}(z) := \mathbb{E}[y|x = z]$ is the Bayes-optimal classifier. In particular, note that $acc(f_{Bayes}) = 100\%$ by construction.

This quantity will be the main object of our analysis, and we will be interested in how it depends on the corruption level $p$ and the choice of pruning strategy $q$, in the infinite-sample limit $N \to \infty$.

For later reference, we also define an empirical version, namely the accuracy of $\widehat{f}_N$ evaluated on the clean dataset $D_N$, namely

$$\widehat{acc}(\widehat{f}_N) := \frac{1}{|M|} |\{i \in M \mid \widehat{f}_N(x_i) = y_i\}|, \tag{13}$$

where $M := \{i \in [N] \mid q_i = 1\}$ collects the indices of training samples which survive pruning by $q$.

### E.3 A Class of Parametrized Pruning Strategies

Given hyper-parameters $\phi_0, \phi_1, \psi_{01}, \psi_{10} \in [0, 1]$, we consider a broad class of parametrized pruning strategies with the following property. For any class labels $k, \ell \in \{0, 1\}$, the random variables $(z_{ik\ell})_{i \in [N]}$ defined by $z_{ik\ell} = 1[y_i = k, y'_i = \ell, q_i = 1]$ are i.i.d. with Bernoulli distribution $Bern(p_{k\ell})$, where

$$
\begin{aligned}
p_{k\ell} &= \mathbb{P}(q_i = 1, y'_i = \ell, y_i = k) \\
&= \mathbb{P}(q_i = 1 \mid y'_i = \ell, y_i = k)\mathbb{P}(y'_i = \ell \mid y_i = k)\mathbb{P}(y_i = k) \\
&= \begin{cases} \phi_k(1-p)/2, & \text{if } k = \ell, \\ \psi_{k\ell}p/2, & \text{else.} \end{cases}
\end{aligned} \tag{14}
$$

and the numbers $p, \phi_k$ and $\psi_{k\ell}$ are defined by

$$p := \mathbb{P}(y'_i \neq y_i), \quad \phi_k = \mathbb{P}(q_i = 1 \mid y'_i = k, y_i = k), \quad \psi_{k\ell} = \mathbb{P}(q_i = 1 \mid y'_i = \ell, y_i = k). \tag{15}$$

Consequently, if $N_{k\ell}$ is the number of training examples that have true label $k$, fake label $\ell$, and survive pruning, then

$$N_{k\ell} := \sum_{i=1}^{N} z_{ik\ell} \tag{16}$$

which is has binomial distribution $Bin(N, p_{k\ell})$. As mentioned in the main text, for simplicity of exposition we considered the following simplifying assumption:

$$\phi_1 = \phi_0 = \phi, \quad \psi_{01} = \psi_{10} = \psi. \tag{17}$$

Such a pruning strategy will be referred to as a verification-pruning strategy with parameter $(\phi, \psi)$.

**Remark E.2.** *Note that the parametrization $(\phi, \psi)$ and $(p_{00}, p_{11})$ describe the same verification-pruning strategy via the following bijective transformation.*

$$p_{00} = p_{11} = (1 - p)\phi/2, \quad p_{01} = p_{10} = p\psi/2. \tag{18}$$

---

[3]Unicity is due to strong convexity of objective function.

### E.4 EXAMPLES

Let us present some notable examples of verification-pruning strategies.

**No Selection.** The case $(\phi, \psi) = (1, 1)$ corresponds to no selection, i.e the entire training dataset is used.

**Oracle Selection.** The case $(\phi, \psi) = (1, 0)$. The selection strategy only keeps indices corresponding to examples in the dataset which have correct label (all corrupted labels discarded).

**Supervised (Margin-Based) Selection.** Let $w_{prune} \in \mathbb{R}^d$, and consider the pruning strategy defined by

$$q_i = 1[y_i'(x_i^\top w_{prune}) > 0].$$

This pruning strategy simplify filters out all examples on which it disagrees on the assigned label.

### E.5 PERFORMANCE BOUNDS FOR MODELS TRAINED WITH VERIFICATION SELECTION

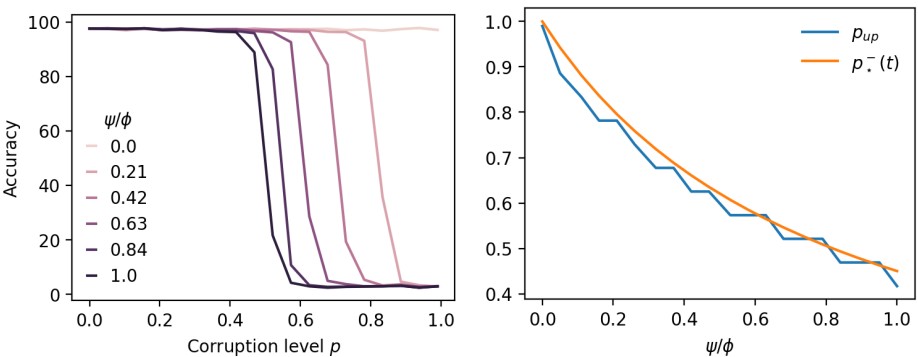

Figure 6: **Empirical Confirmation of Theorem E.3.** Comparing the breakdown points of different models. Here, the task is classifying a Gaussian mixture, with infinite training samples from datasets generated from a model with classification error rate $p$ (x-axis). Notice the sharp phrase-transitions where the model suddenly switches from perfect accuracy to worse-than-chance, a phenomenon predicted by Theorem E.3. **Left.** Performance of verification-pruning strategies with different values of the hyper-parameters $(\phi, \psi)$. Recall that the case $\psi/\phi = 1$ corresponds to no pruning, while $\psi/\phi = 0$ corresponds to oracle selection. **Right.** Comparing $p_{up}$, approximated with $\sup\{p \mid acc(\widehat{f}_N) \geq 90\%\}$ (computed empirically), against the analytic estimate $p_\star^-(t)$ given in Theorem E.3 (for $t = 0.1$). Again, the results are in excellent agreement with the predictions of the theorem.

The following is one of our main results (proved in Appendix F).

**Theorem E.3.** *Suppose Condition E.1 is in order. Fix $\phi, \psi, t \in (0, 1)$ and define $p_\star^\pm(t) \in (0, 1)$ by*

$$p_\star^-(t) := \frac{1 - t}{1 - t + (1 + t)\psi/\phi}, \quad p_\star^+ := \frac{1 + t}{1 + t + (1 - t)\psi/\phi} \tag{19}$$

*If $p < p_\star^-(t)$, then the limit $N \to \infty$ it holds w.p $1 - o(1)$ that the $acc(\widehat{f}_N) = 100\%$ for a downstream model $\widehat{f}_N$ trained on data from a generator with error rate $p$ pruned with a verification-pruning strategy with parameters $(\phi, \psi)$.*

*On the other hand, if $p > p_\star^+$, then in the limit $N \to \infty$ it holds w.p $1 - o(1)$ that the $acc(\widehat{f}_N) = 0\%$ for a downstream model $\widehat{f}_N$.*

*Thus, there is a sharp phase-transition around the corruption level $p_\star := 1/(1 + \psi/\phi)$: as $p$ is increased past level $p_\star$, the downstream model $\widehat{f}_N$ abruptly switches from being perfectly accurate, to perfectly inaccurate!*

See Figure 6 for an empirical illustration of the theorem.

The thresholds $p_\star^\pm(t)$ appearing in the above theorem are proxies for the so-called *breakdown points* $p_{up} \geq p_{down}$ defined by

$$p_{up} = \inf\left\{ p \in [0,1] \,\big|\, acc(\widehat{f}_N) \overset{a.s}{\to} 0\% \text{ in the limit } N \to \infty \right\}, \tag{20}$$

$$p_{down} = \sup\left\{ p \in [0,1] \,\big|\, acc(\widehat{f}_N) \overset{a.s}{\to} 100\% \text{ in the limit } N \to \infty \right\}. \tag{21}$$

Theorem E.3 implies $p_{down} \geq p_\star^-(t)$ and $p_{up} \leq p_\star^+(t)$ for all $t \in (0,1)$. Consequently,

**Corollary E.4.** *Under the hypotheses of Theorem E.3, it holds that $p_{up} = p_{down}$.*

### E.6    SOME CONSEQUENCES OF THEOREM E.3

We now present some illustrious applications of Theorem E.3. These examples are empirically confirmed in Figure 6.

**No Selection.**    Here, we have $\psi/\phi = 1$ and so the downstream model achieves $100\%$ accuracy for all values of corruption parameter $p$ up to the proxy breakdown point predicted by Theorem E.3 is then $p_\star^- = 1/2 - t/2$.

**Oracle Selection.**    For this scenario, $\psi/\phi = 0$ and so Theorem E.3 predicts that the downstream model $\widehat{f}_N$ achieves $100\%$ accuracy for all values of corruption parameter $p$ up to the breakdown point $p_\star^- = 1$. This is perhaps not so surprising in hindsight. The point is that even for moderately large values of $\psi/\phi$, the proxy breakdown point $p_\star^-$ given in equation 19 can still be quite close to 1.

**Self-supervised (Margin-Based) Selection.**    Consider Gaussian mixture data with means $\pm\mu$, and consider a margin-based pruning strategy in Equation (5). It is clear that $\phi$ and $\psi$ only depend on all the 3 angles between the set of vectors $\{w_*, w_{gen}, w_{prune}\}$, with $w_* = \mu$.

**Supervised Selection.**    Consider isotropic Gaussian mixture data with means $\pm\mu$, and a pruning strategy as in Eq. (5). The parameters $(\phi, \psi)$ only depend on the angles $\theta_{gen}, \theta_{prune}, \theta \in [0,\pi]$ given by

$$\begin{aligned} \theta_{gen} &:= \angle(w_{gen}, \mu),\ \theta_{prune} := \angle(w_{prune}, \mu), \\ \theta &:= \angle(w_{prune}, w_{gen}). \end{aligned} \tag{22}$$

Conditioned on $y = 1$ and using $\mathbb{E}\|x\|^2 = \|\mu\|_2^2 + \operatorname{tr}\Sigma = 1$ (and $\|\mu\|_2 < 1$), we get $x \sim \mathcal{N}(\mu, \frac{1}{(1-\|\mu\|_2^2)d}\mathbf{I}_d)$. We can rewrite $x$ as:

$$x = \mu + \frac{1}{(1 - \|\mu\|_2^2)d}\eta, \tag{23}$$

where $\eta \sim \mathcal{N}(0, \mathbf{I}_d)$. For simplicity of calculation, let us further assume here that $\|w_{gen}\|_2^2 = \|w_{prune}\|_2^2 = 1$.

$$\begin{aligned} \phi_1 &= \mathbb{P}(x^\top w_{prune} > 0 \mid y = 1, y' = 1) \\ &= \frac{\mathbb{P}(x^\top w_{prune} > 0, y = 1, x^\top w_{gen} > 0)}{\mathbb{P}(y = 1, x^\top w_{gen} > 0)} \\ &= \frac{\mathbb{P}(\eta^\top w_{prune} > -\|\mu\|_2 d(1 - \|\mu\|_2^2)\cos\theta_{prune}, \eta^\top w_{gen} > -\|\mu\|_2 d(1 - \|\mu\|_2^2)\cos\theta_{gen})}{\mathbb{P}(\eta^\top w_{gen} > -\|\mu\|_2 d(1 - \|\mu\|_2^2)\cos\theta_{gen})}. \end{aligned}$$

In the second step, we use the definition of conditional probability and use equation 23 from line 2 to line 3. The random variables $\eta^\top w_{gen}$ and $\eta^\top w_{prune}$ are jointly Gaussian with:

$$\begin{pmatrix} \eta^\top w_{gen} \\ \eta^\top w_{prune} \end{pmatrix} \sim \mathcal{N}\left(\mathbf{0}, \begin{pmatrix} 1 & \cos\theta \\ \cos\theta & 1 \end{pmatrix}\right).$$

Let $\Phi$ be the CDF of the standard normal distribution and $\Phi_2$ the CDF of the bivariate normal distribution, where $\Phi_2(x, y; \rho)$ is defined as:

$$\Phi_2(x, y; \rho) := \mathbb{P}(X \leq x, Y \leq y)$$

for $(X, Y) \sim \mathcal{N}\left(\mathbf{0}, \begin{pmatrix} 1 & \rho \\ \rho & 1 \end{pmatrix}\right)$. Denote $c_1 = -\|\mu\|_2 d(1 - \|\mu\|_2^2) \cos \theta_{\mathrm{prune}}$ and $c_2 = -\|\mu\|_2 d(1 - \|\mu\|_2^2) \cos \theta_{\mathrm{gen}}$. We have:

$$\phi_1 = \frac{1 - \Phi(c_1) - \Phi(c_2) + \Phi_2(c_1, c_2; \cos \theta)}{1 - \Phi(c_2)}.$$

All the distributions are symmetric, and we have $\phi_0 = \phi_1 = \phi$. In the same spirit, $\psi = \psi_{10} = \psi_{01}$, with

$$\psi = \frac{\Phi(c_1) + \Phi(c_2) - \Phi_2(c_1, c_2; \cos \theta)}{\Phi(c_2)}.$$

### E.7 SKETCH OF PROOF OF THEOREM E.3

The proof is based on the following representation (refer to Proposition F.2) of the accuracy of the downstream classifier $\widehat{f}_N$ evaluated on the the clean training dataset $D_N$, namely

$$\widehat{acc}(\widehat{f}_N) = \frac{N_{11}\mathbf{1}_{\overline{A}<1/2} + N_{00}\mathbf{1}_{\overline{D}<1/2} + N_{10}\mathbf{1}_{\overline{B}>1/2} + N_{01}\mathbf{1}_{\overline{C}>1/2}}{N_{11} + N_{00} + N_{10} + N_{01}}, \tag{24}$$

for some random some random variables $\overline{A}, \overline{B}, \overline{C}, \overline{D} \in (0, 1)$ which depend on the $N_{k\ell}$'s defined in equation 16.

**Remark E.5.** *We only compute the accuracy $\widehat{acc}(\widehat{f}_N)$ of the downstream model $\widehat{f}_N$ evaluated on the clean training dataset $D_N$. By classical results in learning theory (McAllester, 2003; Shalev-Shwartz & Ben-David, 2014; Kakade et al., 2008), we know that the gap to the population version (test accuracy) $acc(\widehat{f}_N)$ shrinks to zero at rate $O(1/\sqrt{N})$, and so since the claim in Theorem E.3 is made only in the limit $N \to \infty$, we are good.*

Next, in Proposition F.3 and Proposition F.4, necessary and sufficient conditions are established to ensure $\overline{A}, \overline{D} < 1/2$ and $\overline{B}, \overline{C} > 1/2$, and therefore $\widehat{acc}(\widehat{f}_N) = 100\%$. These conditions are given explicitly in terms of the $N_{k\ell}$'s. Finally, in Proposition F.5, concentration of measure is used to control the $N_{k\ell}$'s, and present the aforementioned conditions in terms of the $p_{k\ell}$'s defined in equation 14, and therefore in terms of $p$, $\phi$, and $\psi$ alone, giving condition equation 19.

## F PROOF OF THEOREM E.3

Our analysis is based on non-trivial extensions of arguments by Das and Sanghavi (2023). Viz,

- We allow for a selection mechanism (aforementioned work does study selection, just self-distillation), and
- We use a careful asymptotic analysis to avoid solving certain complicated fixed-point equations defining the weights vector $\widehat{w}_N$ of the downstream model $\widehat{f}_N$.

### F.1 PRELIMINARY COMPUTATIONS

For later use, given a pruning strategy $q$, define the following objects

$$I_k := \{j \in [N] \mid y_j = k\}, \tag{25}$$

$$I'_\ell := \{j \in [n] \mid y'_j = \ell\}, \tag{26}$$

$$M := \{i \in [N] \mid q_i = 1\}, \tag{27}$$

$$N_{k\ell} := \sum_{i \in I_k \cap I'_\ell} q_i = |I_k \cap I'_\ell \cap M|, \tag{28}$$

$$R := 1 - a > 0. \tag{29}$$

Thus, $N_{k\ell}$ is the number of training examples that have true label $k$, fake label $\ell$, and survive pruning. The following result will be crucial in the sequel.

**Proposition F.1.** *We have the representation* $\widehat{w} = \sum_{i \in M} \alpha_i x_i$, *where*

$$\alpha_i = \begin{cases} A, & \text{if } i \in I_1 \cap I_1' \cap M, \\ -B, & \text{if } i \in I_1 \cap I_0' \cap M, \\ C, & \text{if } i \in I_0 \cap I_1' \cap M, \\ -D, & \text{if } i \in I_0 \cap I_0' \cap M, \end{cases} \tag{30}$$

*and* $A, B, C, D \geq 0$ *solve the following system of equations*

$$\begin{aligned} \gamma A &= \sigma(-(aN_{11}A - aN_{10}B + bN_{01}C - bN_{00}D) - RA), \\ \gamma B &= \sigma(aN_{11}A - aN_{10}B + bN_{01}C - bN_{00}D - RB), \\ \gamma C &= \sigma(-(bN_{11}A - bN_{10}B + aN_{01}C - aN_{00}D) - RC), \\ \gamma D &= \sigma(bN_{11}A - bN_{10}B + aN_{01}C - aN_{00}D - RD). \end{aligned} \tag{31}$$

*Proof.* The following result is inspired by Das & Sanghavi (2023) and the proof is similar. Observe that KKT conditions $\nabla L(w) = 0$ give $\sum_{i=1}^N q_i(\hat{y}_i - y_i')x_i + \gamma w = 0$, i.e

$$w = \sum_{i=1}^N q_i \alpha_i x_i, \text{ with } \alpha_i := \frac{y_i' - \hat{y}_i}{\gamma}, \ \hat{y}_i := \sigma(v_i), \ v_i = x_i^\top w. \tag{32}$$

One then computes

$$\begin{aligned} v_i = x_i^\top w &= \sum_{j=1}^N q_i \alpha_i x_i^\top x_j = q_i \alpha_i + \begin{cases} a(s - q_i\alpha_i) + bt, & \text{if } i \in I_1, \\ a(t - q_i\alpha_i) + bs, & \text{if } i \in I_0, \end{cases} \\ &= \begin{cases} as + bt + Rq_i\alpha_i, & \text{if } i \in I_1, \\ bs + at + Rq_i\alpha_i, & \text{if } i \in I_0, \end{cases} \end{aligned} \tag{33}$$

where $s \geq 0$ and $t \geq 0$ are given by

$$s := \sum_{j \in I_1} q_j \alpha_j, \quad t := \sum_{i \in I_0} q_j \alpha_j. \tag{34}$$

We deduce that for any $i \in M$,

$$\gamma \alpha_i = y_i' - \sigma(v_i) = \begin{cases} 1 - \sigma(as + bt + Rq_i\alpha_i), & \text{if } i \in I_1 \cap I_1', \\ -\sigma(as + bt + Rq_i\alpha_i), & \text{if } i \in I_1 \cap I_0', \\ 1 - \sigma(bs + at + Rq_i\alpha_i), & \text{if } i \in I_0 \cap I_1', \\ -\sigma(bs + at + Rq_i\alpha_i), & \text{if } i \in I_0 \cap I_0'. \end{cases} \tag{35}$$

Due to monotonicity of $\sigma$, we deduce the existence of $A, B, C, D \geq 0$ such that

$$\alpha_i = \begin{cases} A, & \text{if } i \in I_1 \cap I_1' \cap M, \\ -B, & \text{if } i \in I_1 \cap I_0' \cap M, \\ C, & \text{if } i \in I_0 \cap I_1' \cap M, \\ -D, & \text{if } i \in I_0 \cap I_0' \cap M. \end{cases} \tag{36}$$

$$\hat{y}_i = y_i' - \gamma\alpha_i = \begin{cases} 1 - \gamma A, & \text{if } i \in I_1 \cap I_1' \cap M, \\ \gamma B, & \text{if } i \in I_1 \cap I_0' \cap M, \\ 1 - \gamma C, & \text{if } i \in I_0 \cap I_1' \cap M, \\ \gamma D, & \text{if } i \in I_0 \cap I_0' \cap M. \end{cases} \tag{37}$$

Furthermore, these scalars must verify

$$\begin{aligned} \gamma A &= 1 - \sigma(as + bt + RA) = \sigma(-(as + bt) - RA), \\ \gamma B &= \sigma(as + bt - RB), \\ \gamma C &= 1 - \sigma(bs + at + RC) = \sigma(-(bs + at) - RC), \\ \gamma D &= \sigma(bs + at - RD). \end{aligned} \tag{38}$$

Finally, observe that,

$$s = N_{11}A - N_{10}B, \quad t = N_{01}C - N_{00}D, \tag{39}$$

from which we get

$$
\begin{aligned}
as + bt &= a(N_{11}A - N_{10}B) + b(N_{01}C - N_{00}D) \\
&= aN_{11}A - aN_{10}B + bN_{01}C - bN_{00}D, \\
bs + at &= b(N_{11}A - N_{10}B) + a(N_{01}C - N_{00}D) \\
&= bN_{11}A - bN_{10}B + aN_{01}C - aN_{00}D.
\end{aligned}
$$

Plugging this into equation 38 gives equation 31. $\qquad\square$

## F.2 ANALYTIC FORMULA FOR ACCURACY EVALUATED CLEAN TRAINING DATA

One computes the accuracy $\widehat{acc}(\widehat{f}_N)$ of the downstream model evaluated on the clean training dataset $D_N$ as

$$\widehat{acc}(\widehat{f}_N) = \frac{1}{|M|} \left( |\{i \in M \mid y_i = 1 \wedge \hat{y}_i > 1/2 \text{ OR } y_i = 0 \wedge \hat{y}_i < 1/2\}| \right).$$

We can rewrite this as follows

$$
\begin{aligned}
|M| \cdot \widehat{acc}(\widehat{f}_N) &= |\{i \in M \mid y_i = 1 \wedge \hat{y}_i > 1/2 \text{ OR } y_i = 0 \wedge \hat{y}_i < 1/2\}| \\
&= |\{i \in I_1 \cap M \mid \hat{y}_i > 1/2\}| + |\{i \in I_0 \cap M \mid \hat{y}_i < 1/2\}| \\
&= \sum_{i \in I_1 \cap M} 1_{\hat{y}_i > 1/2} + \sum_{i \in I_0 \cap M} 1_{\hat{y}_i < 1/2} \\
&= |I_1 \cap I_1' \cap M| 1_{\gamma A < 1/2} + |I_1 \cap I_0' \cap M| 1_{\gamma B > 1/2} \\
&\quad + |I_0 \cap I_1' \cap M| 1_{\gamma C > 1/2} + |I_0 \cap I_0' \cap M| 1_{\gamma D < 1/2} \\
&= N_{11} 1_{\gamma A < 1/2} + N_{00} 1_{\gamma D < 1/2} + N_{10} 1_{\gamma B > 1/2} + N_{01} A_{\gamma C > 1/2}.
\end{aligned}
\tag{40}
$$

On the other hand, it is clear that the size of the mask is $|M| = \sum_{k,\ell} N_{k\ell}$. Putting things together gives the following result which shall be crucial in the sequel.

**Proposition F.2.** *For any $\phi, \psi \in [0, 1]$, there is a solution $(A, B, C, D)$ of the system of equations equation 31 such that*

$$\widehat{acc}(\widehat{f}_N) = \frac{N_{11} 1_{\overline{A} < 1/2} + N_{00} 1_{\overline{D} < 1/2} + N_{10} 1_{\overline{B} > 1/2} + N_{01} 1_{\overline{C} > 1/2}}{N_{11} + N_{00} + N_{10} + N_{01}}, \tag{41}$$

*where $\overline{A} := \gamma A$, $\overline{B} = \gamma B$, $\overline{C} = \gamma C$, and $\overline{D} = \gamma D$ as usual.*

Thus, to attain 100% accuracy, it suffices to have $\overline{A}, \overline{D} < 1/2$ and $\overline{B}, \overline{C} > 1/2$. The proof of Theorem E.3 will be all about establishing sufficient conditions which ensure these inequalities.

## F.3 SUFFICIENT CONDITIONS FOR PERFECT ACCURACY

Note that since $\gamma = N\lambda$ with $\lambda > 0$ fixed and $N \to \infty$, we have $\gamma \to \infty$ and system of equations equation 31 simplify to[4]

$$
\begin{aligned}
\overline{B} &= \sigma((aN_{11}\overline{A} - aN_{10}\overline{B} + bN_{01}\overline{C} - bN_{00}\overline{D})/\gamma), \\
\overline{A} &= \sigma(-(aN_{11}\overline{A} - aN_{10}\overline{B} + bN_{01}\overline{C} - bN_{00}\overline{D})/\gamma) = 1 - \overline{B}, \\
\overline{D} &= \sigma((bN_{11}\overline{A} - bN_{10}\overline{B} + aN_{01}\overline{C} - aN_{00}\overline{D})/\gamma), \\
\overline{C} &= \sigma(-(bN_{11}\overline{A} - bN_{10}\overline{B} + aN_{01}\overline{C} - aN_{00}\overline{D})/\gamma) = 1 - \overline{D},
\end{aligned}
\tag{42}
$$

where $\overline{A} := \gamma A$, $\overline{B} = \gamma B$, $\overline{C} = \gamma C$, $\overline{D} = \gamma D$ as usual, and we have used the elementary property that $\sigma(-z) = 1 - \sigma(z)$. Eliminating $\overline{A}$ and $\overline{C}$, the above equations further collapse to

$$
\begin{aligned}
\overline{B} &= \sigma((aN_{11}(1 - \overline{B}) - aN_{10}\overline{B} + bN_{01}(1 - \overline{D}) - bN_{00}\overline{D})/\gamma), \\
&= \sigma((aN_{11} + bN_{01} - a(N_{11} + N_{10})\overline{B} - b(N_{01} + N_{00})\overline{D})/\gamma), \\
\overline{D} &= \sigma((bN_{11}(1 - \overline{B}) - bN_{10}\overline{B} + aN_{01}(1 - \overline{D}) - aN_{00}\overline{D})/\gamma) \\
&= \sigma((bN_{11} + aN_{01} - b(N_{10} + N_{10})\overline{B} - a(N_{01} + N_{00})\overline{D})/\gamma).
\end{aligned}
\tag{43}
$$

---

[4]These simplifications are made possible by the *Mean Value Theorem*.

Two special cases are tractable.

**The Symmetric Case:** $b = -a$. We have $\overline{D} = 1 - \overline{B}$, and thus the equations become

$$\overline{D} = \overline{A}, \quad \overline{C} = \overline{B}, \quad \overline{D} = 1 - \overline{B},$$
$$\overline{B} = \sigma((a(N_{11} + N_{00}) - a(N_{11} + N_{10} + N_{01} + N_{00})\overline{B})/\gamma). \tag{44}$$

If $\overline{B} \leq 1/2$, then we must have $\overline{B} \geq (N_{11} + N_{00})/(N_{11} + N_{10} + N_{01} + N_{00})$, which is impossible if we impose

$$N_{10} + N_{01} < N_{11} + N_{00}, \tag{45}$$

i.e the number of bad indices which survive is smaller than the number of good indices which survive pruning. Thus, under the previous condition, we must have $\overline{C} = \overline{B} > 1/2$ and $\overline{A} = \overline{D} = 1 - \overline{B} < 1/2$. By symmetry of the preceeding argument we know that the condition is also necessary. We deduce the following result.

**Proposition F.3.** *Suppose $b = -a$. Then, for any solution $(A, B, C, D)$ of the system of equations equation 31, the inequalities*

$$\overline{C} = \overline{B} > 1/2, \quad \overline{D} = \overline{A} < 1/2, \tag{46}$$

*hold if and only iff $N_{10} + N_{01} < N_{11} + N_{00}$.*

**Skewed Case:** $b = 0$. Here, we have

$$\overline{B} = \sigma(a(N_{11} - (N_{11} + N_{10})\overline{B})/\gamma),$$
$$\overline{D} = \sigma(a(N_{01} - (N_{01} + N_{00})\overline{D})/\gamma) \tag{47}$$

If $\overline{B} \leq 1/2$, then $\overline{B} \geq N_{11}/(N_{11} + N_{10})$, which is impossible if we impose

$$N_{10} < N_{11}, \tag{48}$$

i.e the number of examples with true label 1, which are incorrectly labelled as 0 in the dataset, which survive pruning is less than the number of examples with true label 1, which are correctly labelled and survive pruning. We deduce that $\overline{B} > 1/2$ under the above condition.

Similarly, if $\overline{D} \geq 1/2$, then $\overline{D} \leq N_{01}/(N_{01} + N_{00})$, which is impossible if we impose

$$N_{01} < N_{00}, \tag{49}$$

i.e the number of with true label 0 but incorrectly labelled as 1 in the dataset, which survive pruning is less than the number of examples with true label 1, which are correctly labelled and survive pruning. We obtain the following result.

**Proposition F.4.** *Suppose $b = 0$. Then, for any solution $(\overline{A}, \overline{B}, \overline{C}, \overline{D})$ of equation 31, we have*

$$\overline{C}, \overline{B} > 1/2 \text{ iff } N_{10} < N_{11}, \tag{50}$$
$$\overline{D}, \overline{A} < 1/2 \text{ iff } N_{01} < N_{00}. \tag{51}$$

### F.4 CONCENTRATION

We shall now derive conditions which are sufficient to ensure the hypothesis in Propositions F.3 and F.4, namely $N_{k\ell} < N_{kk}$ for all $k, \ell \in \{0, 1\}$ with $k \neq \ell$. Recall that for any $k, \ell \in \{0, 1\}$, the counter $N_{k\ell}$ is random with binomial distribution $Bin(N, p_{k\ell})$. Now, by basic binomial concentration, we know that if $p, \psi \in [0, 1)$ and $\phi \in (0, 1]$, then for any fixed $t \in (0, 1)$, it holds w.p $1 - o(1)$ that

$$\begin{cases} N_{k\ell} \leq (1 + t)Np_{k\ell}, & \text{if } k \neq \ell, \\ N_{k\ell} \geq (1 - t)Np_{k\ell}, & \text{if } k = \ell. \end{cases} \tag{52}$$

In particular, w.p $1 - o(1)$, it holds that

$$N_{k\ell} \leq (1 + t)Np_{k\ell}, \tag{53}$$
$$N_{kk} \geq (1 - t)Np_{kk}. \tag{54}$$

Comparing the above inequalities, we deduce the following result.

**Proposition F.5.** *If the following condition holds*

$$\frac{p_{01} + p_{10}}{p_{00} + p_{11}} < \frac{1-t}{1+t} = 1 - \epsilon \text{ with } \epsilon := \frac{2t}{1+t}, \tag{55}$$

*then w.p* $1 - o(1)$ *it holds that*

$$N_{10} + N_{01} < N_{11} + N_{00}. \tag{56}$$

### F.5 PROOF OF THEOREM E.3

Follows directly from putting together Propositions F.2, F.3, F.4, and F.5, and then solving the inequality

$$\frac{1-t}{1+t} \leq \frac{p_{01} + p_{10}}{p_{00} + p_{11}} = \frac{2p\psi}{2(1-p)\phi} = \frac{p\psi}{(1-p)\phi}$$

for $p$. $\qquad\square$

