# OpenReview forum: "Beyond Model Collapse: Scaling Up with Synthesized Data Requires Verification"
_ICLR.cc/2025/Conference — ICLR 2025 Poster_

### Official Review · Reviewer_rFSV · 2024-10-28

**Soundness:** 4
**Presentation:** 4
**Contribution:** 2
**Rating:** 6
**Confidence:** 3

**Summary:**

The authors study whether filtering synthetic data (collected from a generator) using verifiers can prevent model collapse. In a binary classification setting, they relate downstream model performance to two quantities: the generator's error $p$ and a quantity related to the verifier's accuracy $p_*$. When the generator's error surpasses the threshold related to the verifier's accuracy ($p > p_*$), model collapse occurs. They then conduct three empirical experiments (binary classification w/ linear models, eigenvalue prediction w/ transformers, and news summarization with Llama-2) in which they show that downstream performance correlates with the verifier performance $p_*$, which they can estimate using some labeled training data.

**Strengths:**

This paper was well-written, flowing through a toy but well-executed theory section (Section 4), and then a set of three nice experiments (Sections 5-6). The grounding in actual experiments on transformers was great; in particular, the news summarization experiment for tasks without exact match notions of correctness improved the generality of this work. I appreciated that the appendices were very thorough, and I was able to find answers to most of my questions by looking through these.

I'm not confident in my familiarity with related work, but I believe that the setting studied here (and the style of analysis, beginning with a toy model and ending with empirical work) is novel and helpful to the synthetic data community.

**Weaknesses:**

*Novelty of results.* From a practitioner's point of view, the conclusion from these results is simple: when it is possible to evaluate a verifier on labeled data, picking a more accurate verifier gives better results on that same task. This seems to be obvious / empirically quite clear in existing work (e.g. Appendix A.1), and I'm not sure if the specific definition of $p_*$ offers gains over just picking by existing accuracy metrics.

**Questions:**

- How might this generalize to language modeling / task-agnostic pretraining, where there is not necessarily a sense of "correctness" for verification?

- Lines 390-395 point out that when using a weak verifier, there's a dependence on $n'$, the amount of synthetic data to train on. This seems practically quite important to establish. Can the authors comment more on how to "take into account the quantity of data selected" (line 395)?

Misc suggestions for a final version:

- I would suggest moving Appendix C into the main text. If necessary for space, I would move Section 3 to the appendix, since these results are reasonably well-known from the test-time compute literature.

- https://arxiv.org/pdf/2209.03942 is a related reference

- Line 365 typo: "and the ground truth $w_*$. 5Having the verified"
- Line 12 typo: "other LLM[s]"

---

> ### Author Response · Authors · 2024-11-20
> **Authors' response**
>
> We thank the reviewer for their thoughtful review and for specifically appreciating our analysis from theory to empirical settings, as well as the quality of our appendix. Below, we provide our responses.
>
> > Novelty of results. From a practitioner's point of view, the conclusion from these results is simple: when it is possible to evaluate a verifier on labeled data, picking a more accurate verifier gives better results on that same task. This seems to be obvious / empirically quite clear in existing work (e.g. Appendix A.1), and I'm not sure if the specific definition of $p_*$ offers gains over just picking by existing accuracy metrics.
>
> Intuitively, the general trend should hold that a more accurate verifier will lead to better results. However, as shown in our news summarization task, also as highlighted by reviewer rwoT and reviewer mXV4, a better model in evaluation metrics does not necessarilly translate into a model more capable of selection. In our setup, evaluting with the same test set and the rouge score, the finetuned Llama-3 model exhibits higher performance compared with the finetuned Llama-2 model. However, when using the Llama-3 model to select the same amount of data compared with using the Llama-2 model, the model trained from the Llama-3 selected set is actually worse than the model trained from the Llama-2 selected set. In this case, an existing accuracy metric does not help select a better model while our $p_*$ metric could correct do so. Specifically, $p_*$ could be estimated with human / experts looking at a set of generated data and how the data servive during selection. It can be estimated without training and thus offers a cheap measure on how the trained model behaves.
>
> > How might this generalize to language modeling / task-agnostic pretraining, where there is not necessarily a sense of "correctness" for verification?
>
> Thank you for raising this important question. We hope our discussion in General Response 1 provides helpful insights.
>
> > Lines 390-395 point out that when using a weak verifier, there's a dependence on $n'$, the amount of synthetic data to train on. This seems practically quite important to establish. Can the authors comment more on how to "take into account the quantity of data selected" (line 395)?
>
> Thanks for raising this point! We hope our General Response 2 address your question.
>
> > suggestions for a final version:
>     1. I would suggest moving Appendix C into the main text. If necessary for space, I would move Section 3 to the appendix, since these results are reasonably well-known from the test-time compute literature.
>     2. https://arxiv.org/pdf/2209.03942 is a related reference
>     3. Line 365 typo: "and the ground truth $w_*$. 5Having the verified"
>     4. Line 12 typo: "other LLM[s]"
>
> Thank you for your valuable suggestions. We have moved Appendix C into the main text before the news summarization and corrected the typos. The paper you sent is of high relevance and we have included it prominently in the introduction. We plan to include a discussion: Model-generated data has also been shown to amplify dataset biases through feedback loops.
>
> Please let us know if our discussion addresses your concerns regarding the novelty of our work. If there are no remaining issues, we kindly request that you consider raising your score.

---

> ### Comment · Reviewer_rFSV · 2024-11-20
>
> Thanks to the authors for the response to the review. The response makes a lot of sense, and I'm satisfied with the quality of the paper. I maintain my original rating.
>
> One final suggestion for the camera-ready: the observation that a more accurate verifier does not necessarily result in better downstream performance has previously been empirically shown in https://arxiv.org/abs/2309.17425. This submission offers some insight as to why; it would be lovely to mention this connection explicitly in the paper.

---

> > ### Author Response · Authors · 2024-11-21
> > **Author Response to Reviewer Comment**
> >
> > Thanks for the further comments.
> >
> > >*One final suggestion for the **camera-ready**: the observation that a more accurate verifier does not necessarily result in better downstream performance has previously been empirically shown in https://arxiv.org/abs/2309.17425. This submission offers some insight as to why; it would be lovely to mention this connection explicitly in the paper.*
> > >
> > We have added a reference to the paper mentioned by the reviewer. See the blue colored text at the end of **Section 6** (page 10 of updated manuscript):
> >
> > "...These observations resonate with the recent work (Fang et al., 2024) where it was also empirically observed that more accurate models might not always lead to better data filters."
> >
> > >*Thanks to the authors for the response to the review. The response makes a lot of sense, and I'm satisfied with the quality of the paper. I maintain my original rating.*
> >
> > We thank the reviewer for acknowledging that our rebuttal addresses all of their concerns. The reviewer also acknowledges being satisfied with the quality of the paper. We equally thank the reviewer for this remark.
> >
> > In light of the above, may we suggest that the reviewer consider increasing their score (which is current 6). Indeed, let us respectfully point out that a rating of 6 is actually not considered good (it is only "marginally above the acceptance threshold", as per the official ICLR definition), and might not be sufficient to get the paper to the **camera-ready** stage (i.e accepted). Thanks in advance for the consideration.

---

### Official Review · Reviewer_rwoT · 2024-11-03

**Soundness:** 3
**Presentation:** 3
**Contribution:** 3
**Rating:** 6
**Confidence:** 3

**Summary:**

The paper presents a verification method to prevent model collapse that happens when models are trained on synthetic data. The proposed proxy $p_{*}$ offers a practical way to estimate the potential performance of models trained on verified synthesized data. Based on it, the synthetic data can be selected to avoid degrading the performance of the model that is trained on mixed real and synthetic data. Tested across three different tasks, the method shows its effectiveness and practical applicability.

**Strengths:**

1. The paper provides a theoretical framework for understanding the role of verification in preventing model collapse.
2. The paper is well-written and easy to follow.
3. The method has been demonstrated effective via empirical experiments in different domains.

**Weaknesses:**

1. Model collapse not only happens when trained on synthetic data but also occurs when synthetic data is involved in the training set iteratively. This paper fails to address model collapse from this dimension.
2. As stated in the paper, Llama 3 performs worse than Llama 2 when as a verifier. Therefore, an advanced model is not necessary to be a good verifier. There could be more discussion on the reason behind it and how to choose verification models.
3. This paper primarily focuses on accuracy for analysis, yet it would be insightful to consider cases where accuracy alone is insufficient for assessing model performance, such as alignment in large language models (LLMs), and explore how the synthetic data selection strategy could be applied in such complex scenarios.

**Questions:**

Figure 5 shows when more selected synthetic data are trained on, both the performance of Oracle and $p_{*}$ become worse, why does this happen? Does it mean synthetic data can still potentially make the model collapse?

---

> ### Author Response · Authors · 2024-11-20
> **Authors' response (1/2)**
>
> We thank the reviewer for the thoughtful evaluation and for recognizing the practical applicability and effectiveness of our proxy $p_*$. Below, we provide our responses to your questions.
>
> > Model collapse not only happens when trained on synthetic data but also occurs when synthetic data is involved in the training set iteratively. This paper fails to address model collapse from this dimension.
>
> Indeed, model collapse can occur across iterations, but we show that it is possible to prevent collapse at each iteration.
>
> In the worst-case scenario, without mixing in real data, [1] demonstrates that using synthetic data generated by the previous model alone causes the subsequent model to perform worse than the generator. Over iterations, this leads to a compounding effect where the model becomes progressively worse.
>
> Our paper addresses this issue by examining each iteration, which consists of one generation and one training process. We demonstrate that with proper verification, the model trained on synthetic data does not perform worse than its generator. When applied iteratively, proper verification ensures that the model does not degrade over time. In fact, the model can improve with each iteration. Thus, our approach effectively mitigates model collapse.
>
> > As stated in the paper, Llama 3 performs worse than Llama 2 when as a verifier. Therefore, an advanced model is not necessary to be a good verifier. There could be more discussion on the reason behind it and how to choose verification models.
>
> The reason can be elaborated even in linear settings. In Equation (6) and Appendix E.6, we introduced three angles between the generator, verifier, and the ground truth, and derived the relationship between $p_*$ and these angles. The theoretical insight is that the performance and $p_*$ depend on all three angles and cannot be reduced solely to the angle between the verifier and the ground truth (i.e., the performance of the verifier).
>
> Intuitively, a verifier may exhibit strong performance but could learn knowledge that is entirely different from the knowledge embedded in the generator. As a result, it may fail as a good selector because it cannot effectively distinguish the responses produced by the generator.
>
> We have included the discussion.
>
> > This paper primarily focuses on accuracy for analysis, yet it would be insightful to consider cases where accuracy alone is insufficient for assessing model performance, such as alignment in large language models (LLMs), and explore how the synthetic data selection strategy could be applied in such complex scenarios.
>
> Thank you for raising this point! We hope we have addressed the question in General Response 1. Additionally, we would like to comment further on alignment in large language models, as we have discussed this topic with Reviewer CePm in connection to RLHF. We believe that verification can play a valuable role in improving optimization for alignment. Verification could be applied both before and after establishing preferences.
>
> For instance, when all sampled generations are equally poor, it may be preferable to discard them entirely. Verification could help improve the generation of responses for exploration and exploitation. Similarly, after preferences are established, response pairs with weak reward signals could be excluded from the DPO/RLHF training data. Verifiers provide a practical advantage in these scenarios by enabling selective filtering, thus preventing low-quality samples from propagating into the reward and policy models.
>
> The main challenge lies in constructing an effective verifier. A potential starting point could involve combining the reference model with the current reward model. However, this remains an open question and requires a completely different analysis for RLHF compared to the Vicuna-style fine-tuning we did. We leave this as a promising direction for future work.

---

> > ### Author Response · Authors · 2024-11-20
> > **Authors' response (2/2)**
> >
> > > Figure 5 shows when more selected synthetic data are trained on, both the performance of Oracle and $p_*$ become worse, why does this happen? Does it mean synthetic data can still potentially make the model collapse?
> >
> > Please allow us to clarify: in the news summarization task, we are constrained by the total number of available news articles. As a result, increasing the selection ratio (i.e., increasing the number of selected data) involves performing selection on the same dataset but with a lowered selection threshold. This introduces more mid-quality samples, which subsequently reduces overall performance. We believe this is why both the performance and $p_*$ degrade.
> >
> > In each sub-figure, the scaling curves represent the scenario of maintaining a fixed selection threshold while increasing the number of samples. In these cases, the curves consistently show an upward trend.
> >
> > In contrast, the math transformer experiment allows us to fix the selection threshold, as we can generate an unlimited number of samples ourselves. The results from this experiment demonstrate that we can avoid model collapse with synthetic data.
> >
> > We would like to thank you once again for reviewing our work. We remain available to address any further questions you may have. We hope our responses address your questions and encourage you to consider raising your evaluation score.

---

### Official Review · Reviewer_mXV4 · 2024-11-03

**Soundness:** 3
**Presentation:** 2
**Contribution:** 3
**Rating:** 6
**Confidence:** 3

**Summary:**

This paper studies a phenomenon known as "model collapse," in which LLMs that are trained on synthetic (LLM-generated) data suffer drops in performance compared to LLMs trained on real data. The authors propose a verification technique for selecting synthetic data, and analyze cases in which this process can improve over collapsed LLMs trained on randomly selected synthetic data. The theoretical analysis is done using a simplified setting of a Gaussian mixture and linear classifiers, and the results are verified empirically in the same setting. Finally, the authors present empirical results using a pretraining setting for eigenvalue prediction and news summarization.

**Strengths:**

This paper covers an important topic, and establishes some of the fundamentals for effectively training on synthetic data -- this is likely going to be an important area of study as we continue to scale foundation models.
- The authors perform a theoretical analysis that characterizes how high-quality the synthetic data needs to be in order for it to be a useful stand-in for real data, i.e., in order to avoid model collapse.
- The authors present practical results for fine-tuning Llama-2 on a news summarization task as well as a pretraining task in which the transformer is trained to predict matrix eigenvalues. For the news summarization task, the authors find that Llama-3 is a worse verifier than Llama-2, which is a peculiar finding.
- For the real data experiments, the authors confirm that the model collapse phenomena actually occurs, and that their verification procedure actually improves over the model collapse baseline.

**Weaknesses:**

- The theoretical analysis could be improved with some form of a finite-sample result. The current result only applies in the infinite sample limit in a simplified setting, although the result is validated empirically (using finitely-many samples, of course).
- The writing in Section 3 can be improved substantially. The purpose of this section and its contribution to the overall narrative are currently unclear, which can be mostly fixed by leading with a crisp motivational statement. More generally, I'm still not sure that I understand the takeaways of this section -- why is it that the model "lacks the inherent capability to autonomously select the best predictions," when using beam search? Is this a statement about decoding, or about the model? Earlier, in the intro, the authors say that models "cannot intrinsically identify the best solution using perplexity alone," yet beam search *is* essentially based on maximizing the likelihood of outputs.
- 4.1 in the theoretical analysis says that the $y'$ variables "[have] been generated by an AI model," but it's unclear what this statement means mathematically. This should be more formal, particularly since it's part of the theoretical analysis.
- Assumption 4.1 introduces "bits" $q_1, ..., q_N$. I assume these represent whether or not a sample has been pruned, however this is not defined until after it's already mentioned, and it is up to the reader to infer what these bits represent.
- The authors compare to "selection with weak supervision," but there is little elaboration about what this actually means. Weak supervision is an overloaded term and refers to many things -- one reading of this could be that the authors refer to programmatic weak supervision, which could make sense because programmatic weak supervision in some sense generates synthetic training data and verifies it via abstain votes.

**Questions:**

There are a few minor writing inconsistencies in the introduction (and more generally, I feel that the writing of the intro could be improved):
- "We validate our theoretical findings in three empirical settings" and "We conduct two large-scale experiments to test our theoretical insights :" are somewhat at odds. And if they aren't, then the writing for this part should be clarified.
- Ranking is only mentioned once, at the very end of the intro. It is unclear how ranking fits into the verification process -- data selection does not necessitate ranking, although ranking is indeed a way to perform data selection. What is meant by this?
- Is verification really all we need? Some form of "verification is all you need" is mentioned throughout the paper, but it seems like there is a long list of things that we need.
- While I imagine that this would be computationally expensive, it would be useful for the authors to include a scaling law analysis under synthetic data, and for that matter, an example of pretraining on synthetic general data. The proposed technique indeed improves over the collapsed baseline, but it does not necessarily approach the performance of training on real data. There are many questions related to scale that I don't think this paper addresses, but I am not going to factor this into my evaluation, as this seems like obtaining an answer to this would be quite expensive. In any case, I would be interested to hear any speculative thoughts that the authors might have about this.

---

> ### Author Response · Authors · 2024-11-20
> **Authors' response (1/2)**
>
> We thank the reviewer for recognizing our analysis as fundamental and for finding our observation that a better model does not necessarily lead to a better verifier to be intriguing. We also appreciate the valuable advice on improving the writing of the introduction. Below are our responses.
>
>
> > The theoretical analysis could be improved with some form of a finite-sample result. The current result only applies in the infinite sample limit in a simplified setting, although the result is validated empirically (using finitely-many samples, of course).
>
> Thank you for highlighting this important extension. We hope our discussion in General Response 2 helps address it.
>
> > The writing in Section 3 can be improved substantially. The purpose of this section and its contribution to the overall narrative are currently unclear, which can be mostly fixed by leading with a crisp motivational statement. More generally, I'm still not sure that I understand the takeaways of this section -- why is it that the model "lacks the inherent capability to autonomously select the best predictions," when using beam search? Is this a statement about decoding, or about the model? Earlier, in the intro, the authors say that models "cannot intrinsically identify the best solution using perplexity alone," yet beam search is essentially based on maximizing the likelihood of outputs.
>
> Thank you for raising this point. Please allow us to clarify this section: the main takeaway is that while the model can generate good samples, as indicated by the accuracy of all the samples in the beams, the sample selected as the "best" based on the lowest perplexity—the model's most confident prediction—is not necessarily of high quality. As the beam size increases, the accuracy of the beam search solution (as chosen by the model) does not improve at all.
>
> From this, we conclude that the model lacks the inherent capability to autonomously select the best predictions for a given question. This underscores that beam search is not an effective method for improving the quality of generated synthetic data. While many tasks benefit from solutions with higher confidence (e.g., greedy decoding often outperforms nucleus sampling with temperature 1), this correlation does not hold for beam search in our setting. High-quality data do exist within the beams, but the model fails to select them effectively.
>
> We hope this clarifies the intent of this section, which we hope demonstrates this point in a clean synthetic data setting that allows for full experimental control.
>
> > 4.1 in the theoretical analysis says that the  y’ variables "[have] been generated by an AI model," but it's unclear what this statement means mathematically. This should be more formal, particularly since it's part of the theoretical analysis.
>
> Thank you for pointing that out. We intended to emphasize that there is no additional assumption beyond that the synthetic labels being generated as independent and identically distributed. We have updated the version to clarify this point.
>
>
> > Assumption 4.1 introduces "bits" $q_1, \cdots, q_N$. I assume these represent whether or not a sample has been pruned, however this is not defined until after it's already mentioned, and it is up to the reader to infer what these bits represent.
>
> The bits are introduced in Section 4.1, "Downstream Model and Pruning," where we first present the pruning strategy and then explain how the downstream models are trained on the pruned dataset. Specifically, we wrote: "We will model our data selection (whether with or without feedback) via a {\em pruning strategy} $q=(q_1, \ldots, q_N)$, where $q_i$ is a bit that indicates whether the $i$th training example from $D_N'$ has survived pruning." This notation is also introduced mathematically in Equation (1), prior to Assumption 4.1.
>
> > The authors compare to "selection with weak supervision," but there is little elaboration about what this actually means. Weak supervision is an overloaded term and refers to many things -- one reading of this could be that the authors refer to programmatic weak supervision, which could make sense because programmatic weak supervision in some sense generates synthetic training data and verifies it via abstain votes.
>
> Thank you for pointing that out. We used "selection with weak supervision" to describe the selection process using the Llama-3 model on the news summarization task, intending to highlight the difference between selecting with a model and selecting with the ROUGE score, which assumes access to the ground truth. However, we agree with the reviewer that the term is overloaded. For clarity, we have updated it to "Llama-3 selection."

---

> ### Author Response · Authors · 2024-11-20
> **Authors' response**
>
> > "We validate our theoretical findings in three empirical settings" and "We conduct two large-scale experiments to test our theoretical insights :" are somewhat at odds. And if they aren't, then the writing for this part should be clarified.
>
> Thank you for raising this point. We used the term "two large-scale experiments" because one of the three experiments involves linear models on Gaussian data, which we did not consider as a large-scale experiment. In the updated version, we have removed the reference to "two" for clarity.
>
>
> > Ranking is only mentioned once, at the very end of the intro. It is unclear how ranking fits into the verification process -- data selection does not necessitate ranking, although ranking is indeed a way to perform data selection. What is meant by this?
>
> We completely agree and have changed *rank* to *distinguish*.
>
> > Is verification really all we need? Some form of "verification is all you need" is mentioned throughout the paper, but it seems like there is a long list of things that we need.
>
> We have removed the expression to enhance formality of the scientific writing.
>
> > While I imagine that this would be computationally expensive, it would be useful for the authors to include a scaling law analysis under synthetic data, and for that matter, an example of pretraining on synthetic general data. The proposed technique indeed improves over the collapsed baseline, but it does not necessarily approach the performance of training on real data. There are many questions related to scale that I don't think this paper addresses, but I am not going to factor this into my evaluation, as this seems like obtaining an answer to this would be quite expensive. In any case, I would be interested to hear any speculative thoughts that the authors might have about this.
>
> We thanks the reviewer for raising such insightful question. We totally agree that scaling is an important factor here which is not fully addressed. We discuss this point in detail in General Response 1.
>
> We would like to thank you once again for reviewing our work. We have further improved the introduction beyond the advice provided. We hope our responses meet your approval and encourage you to consider raising your score, if so.

---

### Official Review · Reviewer_CePm · 2024-11-04

**Soundness:** 3
**Presentation:** 3
**Contribution:** 4
**Rating:** 8
**Confidence:** 4

**Summary:**

The work investigates synthetic data in LLM training and concerns about model collapse (catastrophic drop in model performance) when LLMs are trained with model-generated labels. Specifically, the work investigates model collapse as a decision problem and utilizes a framework where a verifier (pruner in the manuscript) vets synthetic examples in an attempt to limit the risk of model collapse. Experiments have demonstrated that this verification-based approach can prevent collapse even when the verifiers are imperfect and are not oracles.

**Strengths:**

- Paper is very well written and easy to understand
- Both synthetic data and model collapse are very important topics in the literature and are of great interest to the audience
- Theoretical derivation is complete and appears to be correct
- Very thorough quantitative experiments with strong results

**Weaknesses:**

- In terms of LLMs, qualitative examples showing model collapse in some of the tasks may benefit the delivery further

**Questions:**

- I am interested in whether the proposed methods can be applied to other modalities (e.g., VLMs, Vision-Language-Action Models in robotics) - many problems face more challenges in data annotation cost and also resort to synthetic data.
- Many LLMs are trained with synthetic data specifically on preferences (e.g., RLHF, DPO). I wonder if parallels can be drawn between the verifiers and a reward model and if model collapse is particularly severe with Vicuna-style finetuning.

---

> ### Author Response · Authors · 2024-11-20
> **Authors' response (1/2)**
>
> Thank you for appreciating our writing, theoretical contributions, and the connection with experimental evidence! We also greatly value the connection with RLHF that you highlighted. Here are our responses.
>
> > In terms of LLMs, qualitative examples showing model collapse in some of the tasks may benefit the delivery further.
>
> Thank you for the suggestion—it significantly improves the delivery! We have included additional qualitative examples from the testset of the news summarization task to compare how the model performs with each selection strategy on the same news (see additions to Appendix D). We specifically highlight the collapse behavior and how selection strategies mitigate it. Here, we provide one example but omit the original news text due to its length.
>
> **Ground truth**: *The International Monetary Fund (IMF) has announced that China's currency, the yuan, will join the fund's basket of reserve currencies.*
>
> **Generation of the model trained without selection**: *2015 will be a big year for the Chinese yuan.*
>
> **Generation of the model trained with Self-selection**: *The International Monetary Fund (IMF) has approved the inclusion of the Chinese yuan in its basket of reserve currencies.*
>
> **Generation of the model trained with Llama-3 selection**: *2015 was a good year for China's currency.*
>
> **Generation of the model trained with oracle selection**: *China's yuan has been added to the International Monetary Fund's basket of reserve currencies, the IMF has announced.*
>
> Notice how the generation without selection collapses: it heavily relies on a clichéd format that lacks specific information. While this approach might work for simple news by piquing curiosity, it fails to convey the depth or significance of the current news. The model trained on generated samples collapses to these formats. The Llama-3 selection does not address this issue, aligning with our results that Llama-3 selection performs poorly despite using a better model.
>
> In contrast, generations with self-selection and oracle selection produce summaries that are specific, informative, and directly address the key event, significantly improving the quality of the output.
>
> > I am interested in whether the proposed methods can be applied to other modalities (e.g., VLMs, Vision-Language-Action Models in robotics) - many problems face more challenges in data annotation cost and also resort to synthetic data.
>
> We completely agree that this is an interesting direction! We hope our General Response 1 provides some tangible insights.
>
> Synthetic data is widely leveraged in multi-modal tasks to tackle challenges such as reducing data annotation costs or augmenting specific modalities. For example, in vision-language models (VLMs), synthetic captions are commonly used to expand datasets for unlabeled images. Similarly, in robotics, synthetic simulations are utilized to train policies for complex action sequences. We believe that implementing proper verification methods can greatly enhance the effectiveness of these approaches. For example, in generating diagnostic reports for medical images, researchers or LLMs might incorrectly use synonym substitution as a form of augmentation, potentially violating domain-specific knowledge. Leveraging LLMs coupled with retrieval-augmented generation (RAG) techniques could help produce a verifier to ensure the accuracy and appropriateness of diagnostic reports.
>
> In settings involving multiple modalities, the theory and the proposed metric $p_*$ may not directly apply, as it primarily captures mappings from prompts to responses. Developing a robust theoretical model for selection in multi-modal tasks and understanding selection mechanisms in this context remain promising directions for future work.
>
> > Many LLMs are trained with synthetic data specifically on preferences (e.g., RLHF, DPO). I wonder if parallels can be drawn between the verifiers and a reward model and if model collapse is particularly severe with Vicuna-style finetuning.
>
> Thank you for the insightful question. We agree that model collapse is particularly severe in Vicuna-style fine-tuning. In this setting, the model is trained directly on generated synthetic data without any external signals. However, in RLHF, generated responses are labeled as 'chosen' or 'rejected' using external signals, which help mitigate collapse in a manner similar to a verifier. Comparing the efficiency and effectiveness of a verifier versus a reward model under the same amount of information or labeling is an intriguing direction for future research.

---

> > ### Author Response · Authors · 2024-11-20
> > **Authors' response (2/2)**
> >
> > Regarding whether the current model can serve as the signal, it appears to perform better in RLHF than as a verifier. We hypothesize that this is due to differences in optimization difficulty. In Vicuna-style fine-tuning, the optimization process is relatively standard. In contrast, RLHF involves additional complexities, such as online versus offline and on-policy versus off-policy optimization, which can hinder the process. As a result, using the model itself as a signal in RLAIF appears to be highly effective [1, 2].
> >
> > Verification and selection can also complement RLHF. In cases where two responses are equally poor or where the reward signal for them is weak, it might be preferable to discard such pairs rather than include them in DPO/RLHF training data. Verifiers offer a practical advantage in these situations by enabling selective filtering, which prevents low-quality samples from propagating into the reward and policy model. We left it for future work.
> >
> > [1] Richard Pang and Yuanzhe Yuan et al. "Iterative reasoning preference optimization." arXiv preprint arXiv:2404.19733 (2024).
> > [2] Tianhao Wu et al. "Meta-rewarding language models: Self-improving alignment with llm-as-a-meta-judge." arXiv preprint arXiv:2407.19594 (2024).
> >
> > Hope these discussion address your questions.

---

> > > ### Comment · Reviewer_CePm · 2024-11-26
> > >
> > > Thank you for the answers. I've updated my score.

---

### Author Response · Authors · 2024-11-20
**General Response: How to do data verification and finite sample (Part 1)**

We thank all the reviewers for their insightful and high-quality feedback. We are glad all reviewers appreciated our theoretical analysis on how verifiers, even imperfect ones, can improve synthetic data, as well as the insights derived from our work—specifically the proxy $p_*$— and found they are well-supported by a comprehensive set of experiments with increasing complexity. We are happy that many reviewers also praised our writing structure and found the observation that a better model is not necessarily a better verifier to be a peculiar finding.

In the following general response, we address some common questions raised in the reviews. A new version incorporating reviewer suggestions has been uploaded, with changes marked in blue.

1. **How can we effectively approach data verification for pretraining when accuracy is not available as a metric, alongside analyzing scaling for pretraining on synthetic general data?** Reviewer rwoT highlighted the challenge of '*cases where accuracy alone is insufficient for assessing model performance*'; Reviewer rFSV inquired about how the method '*generalizes to language modeling or task-agnostic pretraining, where there is not necessarily a sense of "correctness" for verification*'; and reviewer mXV4 raised the question of conducting '*a scaling law analysis under synthetic data, and for that matter, an example of pretraining on synthetic general data*'; Additionally, Reviewer CePm touched on a related issue, suggesting the extension of verification methods to other modalities.

These are excellent questions and remarks following up on our work. Indeed, the challenging aspect for general settings lies in effectively building and leveraging verifiers to select high-quality data. First of all, verifiers can encompass a combination of metrics and criteria tailored to specific domains, often extending beyond accuracy. Any correlated feedback signals can be used to guide data selection, including the concept of "programmatic weak supervision" mentioned by Reviewer mXV4. In the context of physical synthetic data, verifiers could include differential equations designed to model the setting; in other scenarios, existing rule-based knowledge can be leveraged. Individual verifiers for each domain can be combined into a mixture of verifiers. To generalize these methods, verifiers could be augmented with LLM acting as processors. Rules, prompts, or even search strategies (retrieval) could be integrated as in-context information, similar to approaches that enhance generated outputs [1, 2].

The most general form of verification could involve leveraging a pretrained general model itself. For instance, the training corpus for Llama-3 was preselected (verified) by Llama-2. This process employs a broader notion of 'correctness', evaluating whether each data point meets the quality threshold necessary for use as pretraining material. Pretrained models acquire some general rules for identifying high-quality data, which can be assessed through metrics like perplexity and other interpretability measures. However, optimizing the use of pretrained models and understanding the extent to which these metrics correlate with data quality remain open areas for further investigation. Recent advancements in enhancing LLMs to act as judges [3,4] underscore their potential for general and alignment tasks, even without explicit metrics. This approach aligns with the intuition presented in our paper that classification tasks are simpler than generation tasks. Furthermore, fine-tuning models specifically for judgment tasks could enhance their effectiveness and utility.

Recent advances in data selection help provide valuable insights. A common approach involves using a high-quality dataset to train a model that generalizes and identifies high-quality data within a larger set [5]. The selected subset can then be used to train an improved model, forming an iterative data engine that can incorporate external information at each stage. Several statistical methods have also been proposed [6, 7], which focus on aligning data with the general distribution in a well-represented feature space without relying on accuracy metrics. For synthetic data specifically, the controllability of the generation process may allow us to tailor these methods in tandem with data generation, resulting in a well-verified dataset.

With these methods, we could test whether the optimal combination achieves scaling performance surpassing that of real data. This would examine whether we could avoid model collapse in the scaling laws, as suggested in [8].

---

> ### Author Response · Authors · 2024-11-20
> **General Response (Part 2)**
>
> Verification of general data is a critical and challenging problem. Recently, startups have emerged with a strong focus on addressing this issue through advanced engineering solutions. Datalogy have demonstrated significant success in data selection with synthetic data incorporated into workflows, particularly in reducing the volume of required data for tasks involving both text and language-label pairs [9]. These promising achievements have the potential to inspire further research and innovation in this direction.
>
> We sincerely thank the reviewer for their insightful questions, which have prompted us to formulate this discussion. The ideas outlined here not only expand on the theoretical framework of our paper but also open avenues for application in diverse domains. We will include this extended discussion in the appendix.
>
> [1] Lewis, Patrick, et al. "Retrieval-augmented generation for knowledge-intensive nlp tasks." Advances in Neural Information Processing Systems 33 (2020): 9459-9474.
>
> [2] Saha, Swarnadeep, et al. "Branch-Solve-Merge Improves Large Language Model Evaluation and Generation." Proceedings of the 2024 Conference of the Association for Computational Linguistics. 2024.
>
> [3] Pang, Richard, et al. "Iterative reasoning preference optimization." arXiv preprint arXiv:2404.19733 (2024).
>
> [4] Wu, Tianhao, et al. "Meta-rewarding language models: Self-improving alignment with llm-as-a-meta-judge." arXiv preprint arXiv:2407.19594 (2024).
>
> [5] Kirillov, Alexander, et al. "Segment anything." Proceedings of the IEEE/CVF International Conference on Computer Vision. 2023.
>
> [6] Wang, Yiping, et al. "Variance Alignment Score: A Simple But Tough-to-Beat Data Selection Method for Multimodal Contrastive Learning." Advances in Neural Information Processing Systems 37 (2024).
>
> [7] Dong, Yijun, et al. "Sketchy Moment Matching: Toward Fast and Provable Data Selection for Finetuning." arXiv preprint arXiv:2407.06120 (2024).
>
> [8] Dohmatob, Elvis, et al. "A Tale of Tails: Model Collapse as a Change of Scaling Laws." Forty-first International Conference on Machine Learning.
>
> [9] https://www.datologyai.com/post/productionized-multimodal-data-curation-at-the-billion-sample-scale
>
> 2. **Extending the results to finite-sample settings.** Reviewer mXV4 suggested that '*the theoretical analysis could be improved with some form of a finite-sample result,*' while Reviewer rFSV emphasized the need to address '*how to take into account the quantity of data selected*'.
>
> Indeed, we limited our theoretical analysis (Theorem 4.2 and Theorem E.3) to the asymptotic regime where the sample size $n$ is cranked to infinity. In this limit, a few important analytical simplifications prevail; the most important being the passage from equations (31) to (42) of the appendix, where we only retain the tail behavior of the sigmoid function, allowing us to obtain fixed-point equations which ultimately lead to our theorem. This asymptotic regime is still rich enough to provide theoretical coverage for our experimental findings.
>
> Though our asymptotic results are indeed experimentally confirmed even in the finite-sample setting, a theoretical analysis of this regime, while still technically possible, will require considerably more work. This will be pursued in a future paper. We can already say that such an analysis would use ideas from random matrix theory and so-called Gaussian comparison inequalities like Gordon's inequality and generalizations thereof.
>
> In summary, we are grateful for the constructive feedback provided by the reviewers. The questions raised are crucial for advancing the effective use of synthetic data in the whole field. We will include these discussions in the appendix and hope that our paper, along with the insights shared here, can serve as a catalyst for further research in this direction.

---

### Meta-Review · Area_Chair_zGpz · 2024-12-22

**Metareview:**

The paper raises an interesting concern that the use of synthetic/LLM-generated data in training causes "model collapse" -- a drop in model performance. Specifically, the authors investigate model collapse as a decision problem with a framework where a verifier vets synthetic examples in an attempt to limit the risk of model collapse. They study whether simply filtering synthetic data (collected from a generator) using verifiers can prevent model collapse. The theoretical analysis is done using a simplified setting of a Gaussian mixture and linear classifiers, and the results are verified empirically in the same setting. Finally, the authors present empirical results using a pretraining setting for eigenvalue prediction and news summarization.

I side with the reviewers who all rated towards acceptance.

**Additional Comments On Reviewer Discussion:**

Unfortunately, half of the reviewers (2 out of 4) never engaged during the discussion period, despite multiple pings from the AC. The other 2 reviewers also engaged rather minimally, although one of them decided to raise the score.

In light of the updates and changes made by the authors, I think the work is significantly improved and grants an acceptance.

---

### Decision · Program_Chairs · 2025-01-22

Accept (Poster)